# Cathepsin-facilitated invasion of BMI1-high hepatocellular carcinoma cells drives bile duct tumor thrombi formation

Lei-Bo Xu [1,2,3,4], Yu-Fei Qin [1,2,3,4], Liangping Su[1,2,4], Cheng Huang[1,2], Qiuping Xu[1,2], Rui Zhang[1,2,3], Xiang-De Shi[1,2,3], Ruipu Sun[1,2], Jiali Chen[1,2], Zhixiao Song [1,2,3], Xue Jiang[1,2], Lihuan Shang[1,2], Gang Xiao[1,2,3], Xiangzhan Kong[1,2], Chao Liu[1,2,3] ✉ & Ping-Pui Wong [1,2] ✉

Bile duct tumor thrombosis (BDTT) is a complication mostly observed in patients with advanced hepatocellular carcinoma (HCC), causing jaundice and associated with poor clinical outcome. However, its underlying molecular mechanism is unclear. Here, we develop spontaneous preclinical HCC animal models with BDTT to identify the role of BMI1 expressing tumor initiating cells (BMI1[high] TICs) in inducing BDTT. BMI1 overexpression transforms liver progenitor cells into BMI1[high] TICs, which possess strong tumorigenicity and increased trans-intrahepatic biliary epithelial migration ability by secreting lysosomal cathepsin B (CTSB). Orthotopic liver implantation of BMI1[high] TICs into mice generates tumors and triggers CTSB mediated bile duct invasion to form tumor thrombus, while CTSB inhibitor treatment prohibits BDTT and extends mouse survival. Clinically, the elevated serum CTSB level determines BDTT incidence in HCC patients. Mechanistically, BMI1 epigenetically up-regulates CTSB secretion in TICs by repressing miR-218-1-3p expression. These findings identify a potential diagnostic and therapeutic target for HCC patients with BDTT.

Bile duct tumor thrombosis (BDTT) is a severe complication that most likely occurred in advanced hepatocellular carcinoma (HCC) patients, which can cause a rapid decline in quality of life (i.e., obstructive jaundice) and associate with worse clinical outcome[1–3]. However, HCC patients with BDTT are often misdiagnosed and mistreated because of its unknown cellular origin and pathogenesis[1,2,4–6]. Although surgical treatment is the only recommended therapy for HCC patients with BDTT, these patients have ~50% probability of recurrence 1 year after surgery, and with a 5-year survival rate below 25%[4,7–9]. Thus, it is a clinically urgent request to develop an effective diagnosis and treatment method for HCC patients accompanied BDTT.

Previous research efforts have mainly focused on exploring the clinical pathology or therapeutic measure of HCC patients with BDTT[7,8,10], while little is known about the developmental process of BDTT. According to a preliminary clinical study, Zhen et al. speculated that highly aggressive tumor cells might invade through the sub-epithelium of the adjacent bile duct wall, leading to the formation of tumor thrombus[10]. Furthermore, they noted an upregulation of BMI1, a member of the polycomb repressive complex 1 associated with tumor cell aggressiveness[11], in a limited subset of HCC patients with BDTT[12]. BMI1 plays a pivotal role in a range of cellular processes, encompassing stem cell self-renewal, proliferation, and the preservation of genomic

[1]Guangdong Provincial Key Laboratory of Malignant Tumor Epigenetics and Gene Regulation, Guangdong-Hong Kong Joint Laboratory for RNA medicine, Sun Yat-sen Memorial Hospital, State Key Laboratory of Oncology in South China, Sun Yat-sen University, Guangzhou 510120, China. [2]Medical Research Center, Sun Yat-sen Memorial Hospital, Sun Yat-sen University, Guangzhou 510120, China. [3]Guangzhou Key Laboratory of Precise Diagnosis and Treatment of Biliary Tract Cancer, Department of Biliary-Pancreatic Surgery, Sun Yat-sen Memorial Hospital, Sun Yat-sen University, Guangzhou 510120, China. [4]These authors contributed equally: Lei-Bo Xu, Yu-Fei Qin, Liangping Su. ✉e-mail: liuchao3@mail.sysu.edu.cn; huangbp3@mail.sysu.edu.cn

stability[13]. Furthermore, it has been implicated in diverse cancers, including hematological malignancies and solid tumors, with its overexpression frequently linked to aggressive tumor behavior, the presence of tumor initiating cells (TICs), and unfavorable prognosis[12,14,15]. At the molecular level, multiple studies have demonstrated that BMI1 can collaborate with the PRC2 protein EZH2, a well-known histone methyltransferase, to suppress gene transcription, likely through the enhancement of trimethylation of H3 at lysine 27 (H3K27me3)[16–18]. This epigenetic modification has been shown to play a crucial role in the regulation of self-renewal in both normal stem cells and TICs, also referred to as cancer stem cells[13,19]. Indeed, Chen et al. showed that BMI1[high] TIC promoted tumor invasive growth and lymph node metastasis in squamous cell carcinoma[14]. Additionally, Wang et al. indicated that BMI1 regulated gastric cancer progression by controlling miR-21 and miR-34a expression[20], while BMI1 itself was capable of repressing miRNA let-7i expression by forming PRC on its regulatory area containing polycomb response elements (PREs)[16]. MiRNAs are known as small non-coding molecules that involve in regulating expression of more than 60% of the human genes[21,22], and their aberrant expression associates with the features of TICs[18,23]. However, the potential involvement of BMI1 and its downstream effector miRNA in BDTT remains unexplored primarily due to the absence of animal models specifically designed to investigate BDTT.

Recent studies have shown that lysosomal cathepsin B (CTSB), a family member of lysosomal cysteine protease, actively modulates the proteolysis of extracellular matrix components, disruption of intracellular communication, degradation of the basement membrane and alteration of cell-cell interactions, thereby it has multiple potential roles in cancer progression[24–28]. Additionally, Venkataraman et al. indicated that CTSB was a potential regulatory target of miRNAs in medulloblastoma[29]. However, it remains unknown whether BMI1 epigenetically regulates CTSB secretion in TICs to modulate BDTT development via miRNA.

Herein, we investigate the cellular origin and molecular mechanism of BDTT in HCC. We provide strong clinical evidence that in HCC patients, elevated BMI1 expression is correlated with increased TIC-related gene signature expression, enhanced BDTT incidence and worse overall survival. Importantly, aberrant expression of BMI1 malignantly transforms liver progenitor cells (LPC) into BMI1[high] TICs, while these cells display high tumorigenicity and enhance trans-intrahepatic biliary epithelial migration/invasion abilities by secreting lysosomal CTSB. Strikingly, orthotopic implantation of BMI1[high] TICs into the livers or spleens of mice forms primary tumors/metastases and undergoes spontaneous BDTT. Further histopathological examination and immunohistochemistry (IHC) analysis of the tumor serial sections demonstrates that BMI1[high] TICs invade into bile ducts from primary liver tumors/metastatic sites to form BDTT. Importantly, treatment with CTSB inhibitor prohibits BDTT and extends survival in orthoptic liver tumor bearing mice independent of primary orthotopic tumor growth. Clinically, the serum CTSB level is dramatically increased in HCC patients with BDTT. Mechanistically, BMI1 epigenetically upregulates CTSB secretion by downregulating miR-218-1-3p expression. Overall, our work identifies the cellular origin and molecular mechanism of BDTT, indicating that harnessing BMI1-miR-213-1-3p-CTSB signaling axis could be a potential therapeutic approach for HCC patients with BDTT.

## Results

### In HCC patients, elevated BMI1 expression is correlated with increased TIC-related protein/gene signature expression, enhanced BDTT incidence and poor prognosis

As the relationship between BMI1 expression and HCC patient prognosis remains underexplored, we therefore performed bioinformatics analysis of the clinical and RNA-sequencing data obtained from the Cancer Genome Atlas (TCGA) database[30,31]. Additionally, we performed IHC examination of our collected HCC patient cohort, indicating that in HCC patients, high expression level of BMI1 correlated with worse overall survival (Fig. 1a and Supplementary Fig. 1a), increased tumor size (Fig. 1b) and elevated cancer progression (Fig. 1c and Supplementary Fig. 1b). We also collected a relatively large sample cohort of HCC patients with or without BDTT in comparison with a previous study[32], and used them to perform an IHC analysis of BMI1 expression, indicating that the patients with BDTT associated strongly with worse overall survival (Fig. 1d), while the patients with BDTT and high BMI1 expression correlated with low percentage of overall survival (Fig. 1e). Additionally, we showed that the elevated BMI1 expression distinguished HCC patients in those with enhanced incidence of BDTT (Fig. 1f).

Given the established role of BMI1 expression in governing TIC self-renewal[13,14], we proceeded to investigate the relationship between the TIC-associated signature (including CD44, CD133, and SOX9)[33,34] and BMI1 expression at both protein and mRNA levels within HCC patients. Our analysis indicated a positive correlation between the expression of the TIC-related protein signature and BMI1 in the tumor regions (Fig. 1g, h). This association was further substantiated through co-immunostaining experiments, affirming a heightened co-expression of BMI1 and CD44/CD133/SOX9 in HCC patients exhibiting high BMI1 expression, in contrast to those with low BMI1 expression (Supplementary Fig. 1c). Additionally, our analysis of the TCGA database[30,31] reaffirmed the correlation between the TIC-related gene signature (comprising CD44, CD133, and SOX9) and BMI1 expression (Supplementary Fig. 1d). Further clinical studies showed that high TIC-related protein/gene signature expression correlated with increased incidence of BDTT (Fig. 1i), poor overall survival (Fig. 1j, k), elevated tumor size (Supplementary Fig. 1e) and enhanced cancer progression (Supplementary Fig. 1f) in HCC patients. Finally, we showed that the HCC patients with high expression of TIC-related protein/gene signature and BMI1 had poor overall survival as compared to the patients with low TIC-related protein/gene signature and BMI1 expression (Fig. 1l, m), whilst the overall survival rate of HCC patients accompanied BDTT and high TIC-related protein signature expression was lower than those patients without BDTT and low TIC-related protein signature expression (Supplementary Fig. 1g). Overall, our clinical studies have revealed a potential association between BMI1 and the expression of TIC-related markers, suggesting a role in the development of BDTT and cancer progression in HCC. This finding has motivated us to delve into its function in regulating cancer cell properties.

### BMI1[high] tumor initiating cells were derived from malignantly transformed liver progenitor cells

To examine the cellular functions of BMI1 in HCC tumorigenesis, we genetically modified BMI1 expression in the rat LPC line WBF344 and human HCC cell lines (i.e., PLC with low endogenous BMI1 level and MHCC97H with high endogenous BMI1) (Fig. 2a, b and Supplementary Fig. 2a). Our results indicated that BMI1 overexpression increased cell growth (Fig. 2c, d), colony formation (Fig. 2e, f), cell migration (Fig. 2g, h) and cell invasion (Fig. 2i, j) as well as tumor sphere formation (Fig. 2k, l) in WB[BMI1] and PLC[BMI1] cells compared to control cells (i.e., empty vector transfected cells, WB[Ctrl]/PLC[Ctrl] cells). Conversely, the depletion of BMI1 using shRNAs or inhibitors in human MHCC97H cells resulted in the opposite effect on these cellular processes (Supplementary Fig. 2a–k).

To determine whether stable BMI1 expressing cells could regulate their invasion ability via paracrine effect, we exposed WB[Ctrl] or PLC[Ctrl] cells to conditioned medium (CM) harvested from either WB[BMI1]/WB[Ctrl] cells or PLC[BMI1]/PLC[Ctrl] cells in a transwell matrigel based invasion system, indicating that exposure with CM from WB[BMI1]/PLC[BMI1] cells increased the invasion in WB[Ctrl] and PLC[Ctrl] cells when compared with the cells exposed to CM from WB[Ctrl] and PLC[Ctrl] cells respectively (Fig. 2m–o). In contrast, exposure of wild type (WT) MHCC97H cells

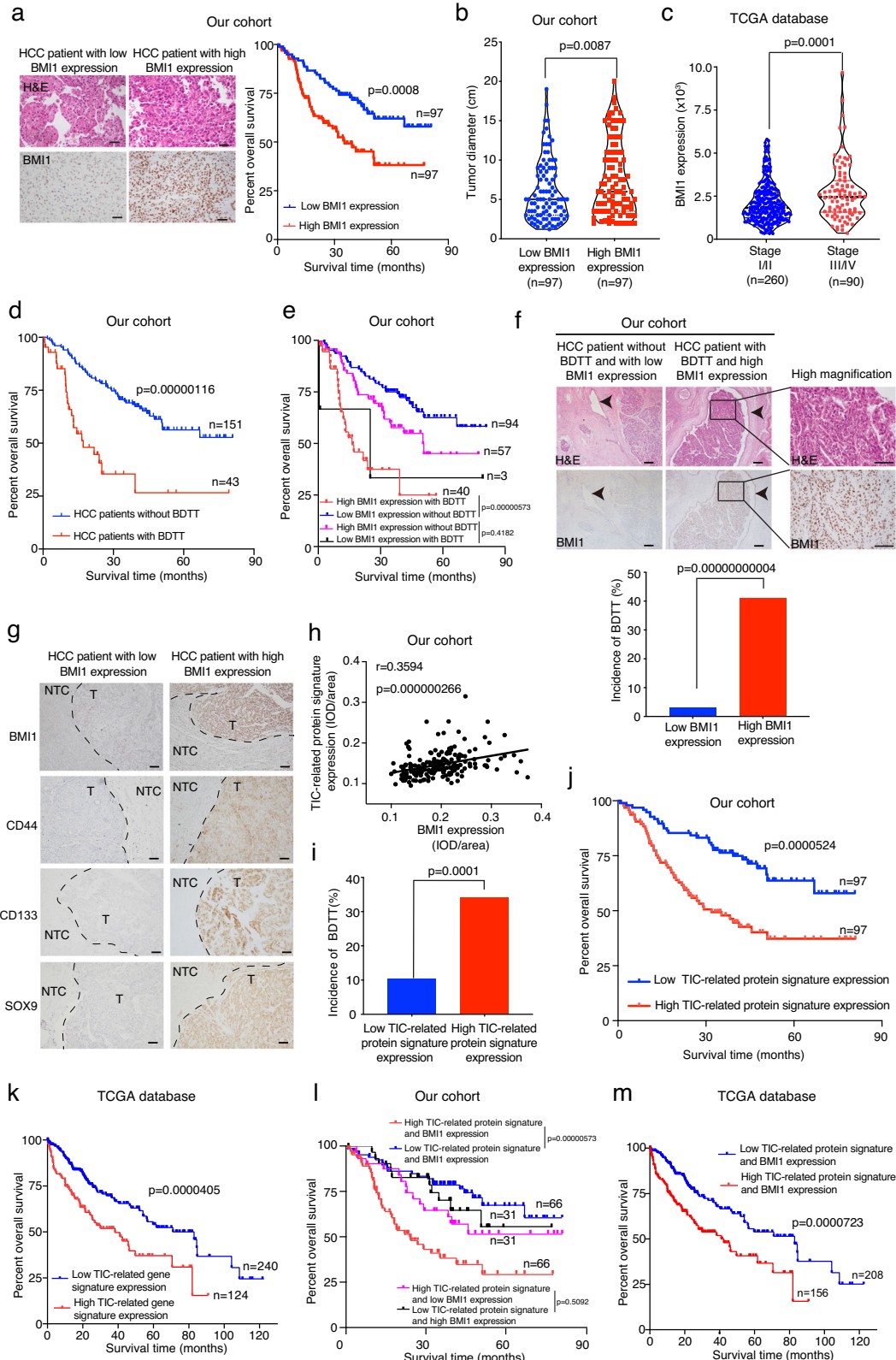

with CM harvested from BMI1 depleted MHCC97H cells or MHCC97H cells pre-treated with BMI1 inhibitors reduced their invasion ability compared to the cells treated with CM derived from either scramble transfected MHCC97H cells or DMSO pre-treated MHCC97H cells (Supplementary Fig. 2l, m). To further examine whether WB$^{BMI1}$ cells possessed the properties of TICs, we performed RT-PCR and western blot analysis of the TIC-related markers, including CD44, CD133 and

SOX9 expression, in WB$^{Ctrl}$ and WB$^{BMI1}$ cells, indicating that BMI1 overexpression upregulated the expression of TIC-related markers in WBF344 cells (Supplementary Fig. 2n, o). Furthermore, we demonstrated that the knockdown of BMI1 expression in human MHCC97H cells resulted in a reduction in the expression of TIC-related markers (Supplementary Fig. 2p). We next examined their tumorigenicity in vivo by injecting small numbers of WB$^{BMI1}$ or WB$^{Ctrl}$ cells (i.e., 5000/

**Fig. 1 | In HCC, high BMI1 expression correlates with enhanced TIC-related gene signature expression, increased incidence of BDTT and worse overall survival.**
**a** Association between BMI1 expression and overall survival in patients with hepatocellular carcinoma ($n = 194$ patients, our cohort). **b** Correlation studies between BMI1 expression and tumor size in HCC patients ($n = 194$ patients, our cohort). **c** Association between BMI1 expression and HCC cancer stages in TCGA (phs000178.v11.p8) ($n = 350$ patients). **d** Kaplan–Meier overall survival study of HCC patients with or without BDTT ($n = 194$ HCC patients, our cohort). **e** Correlation between BMI1 expression and overall survival in HCC patients with or without BDTT ($n = 194$ HCC patients, our cohort). **f** In HCC patients, high BMI1 expression associated with increased incidence of BDTT. Representative pictures of H&E and BMI1 immunohistochemical staining in tumors derived from each group are given. Arrow indicates the position of bile duct. Magnified pictures from each group are given. **g, h** The relationship between TIC-related protein signature (including CD44/CD133/SOX9) and BMI1 expression in HCC patients. Dotted lines indicate the borders between tumor (T) and the non-tumor compartments (NTC). **i,**

**j** Correlation studies between TIC-related protein signature expression and BDTT incidence/overall survival in HCC patients ($n = 194$ patients, our cohort).
**k** Association between TIC-related gene signature expression and HCC patient overall survival in TCGA (phs000178.v11.p8) ($n = 364$ patients). **l** Elevated expression of the TIC-related protein signature along with BMI1 was associated with a more unfavorable patient survival outcome when contrasted with patients exhibiting low TIC-related gene signature and BMI1 expression. However, no distinction in overall survival emerged between HCC patients with either low TIC-related gene signature and high BMI1 expression or those displaying high TIC-related gene signature and low BMI1 expression ($n = 194$ HCC patients, our cohort).
**m** Kaplan–Meier survival study of TIC-related gene signature and BMI1 expression in HCC patients from TCGA (phs000178.v11.p8) ($n = 364$ patients). **a**, **d**, **e**, **j**, **k**–**m** Log-rank (Mantel-Cox) test. **b**, **c** Unpaired two-tailed $t$ test. **h** Two-sided Pearson correlation test. **f**, **i** Two-sided Chi-square test. Scale bars in (**a**, **f**, **g**) 100 μm. Source data are provided in a Source data file.

---

10,000 cells) into nude mice subcutaneously, indicating that the injection of 5000 WB$^{BMI1}$ cells was sufficient to form subcutaneous tumors in nude mice with 100% efficacy. In contrast, the WB$^{Ctrl}$ cells failed to generate any palpable tumor (Fig. 2p, q). Overall, these findings demonstrate that the overexpression of BMI1 can induce a malignant transformation of LPC into BMI1$^{high}$ TICs, referred to as WB$^{BMI1}$.

## Orthotopic injection of WB$^{BMI1}$ cells into the liver/spleen of mice can generate primary tumors/metastases and undergo spontaneous bile duct tumor thrombosis

To examine whether BMI1$^{high}$ TICs could initiate orthotopic tumor formation and BDTT, we performed an orthotopic implantation of WB$^{BMI1}$ cells into the Glisson 's capsules of the livers in nude mice (Fig. 3a and Supplementary Fig. 3a), indicating that 40% of the mice implanted with WB$^{BMI1}$ cells developed jaundice 4 weeks after the injection (Fig. 3b). Notably, the appearance of jaundice was one of the clinical performances in HCC patients with BDTT[2]. Additionally, liver function blood test indicated that the direct and total bilirubin level was up-regulated in WB$^{BMI1}$ implanted mice with jaundice as compared with either WB$^{Ctrl}$ implanted mice or WB$^{BMI1}$ implanted mice without jaundice (Supplementary Table 3). Magnetic resonance imaging (MRI) showed that orthotopic implantation of the WB$^{BMI1}$ cells into the Glisson's capsules of livers in nude mice generated liver tumors in 4 weeks, while the WB$^{Ctrl}$ implanted cells failed to form any tumor (Fig. 3c). We also showed that the WB$^{BMI1}$ implanted cells generated primary liver tumors at 100% incidence, while 40% of the WB$^{BMI1}$ tumor bearing mice developed gallbladder and bile duct enlargement (Fig. 3d and Supplementary Fig. 3b). In contrast, none of the WB$^{Ctrl}$ implanted mice showed any of these phenotypes (Fig. 3c, d). Further histopathological study of the tumor serial sections indicated the presence of bile duct tumor thrombi in WB$^{BMI1}$ orthotopic tumor bearing mice with jaundice, but not in the mice implanted with WB$^{Ctrl}$ cells (Fig. 3e). Importantly, we showed that the tumor thrombi were arisen from the invasion of WB$^{BMI1}$ cells from primary tumor sites into the lumen of bile ducts (Fig. 3e). Since cytokeratin 7 (CK7) and cytokeratin 19 (CK19) is generally expressed in bile ductal epithelium[35,36], we therefore carried out IHC staining of CK7 and CK9 in liver tissue/tumor sections derived from the mice implanted with either WB$^{Ctrl}$ or WB$^{BMI1}$ cells, indicating the presence of CK7 and CK19 positive bile duct epithelial fragments on the surface of the tumor thrombi in WB$^{BMI1}$ implanted mice with jaundice (Fig. 3e). Moreover, our co-immunostaining experiments have furnished supplementary evidence showcasing the infiltration of BMI1 positive cells (WB$^{BMI1}$) through the CK7 positive bile ducts, ultimately resulting in the formation of tumor thrombi within the bile duct lumen (Supplementary Fig. 3c). To ascertain whether BMI1 expressing TICs (WB$^{BMI1}$) exhibited invasive behavior within the bile ducts, we also conducted orthotopic injections of GFP overexpressing WB$^{Ctrl}$ and

WB$^{BMI1}$ cells into the Glisson's capsules of livers in nude mice respectively. Subsequently, we harvested orthotopically developed liver tumors in mice implanted with WB$^{BMI1}$ cells that developed jaundice, as well as the livers of mice implanted with WB$^{Ctrl}$ cells, for further co-immunostaining analysis involving GFP and CK7. The results from these analyses consistently demonstrated the effective infiltration of GFP overexpressing WB$^{BMI1}$ cells through the layer of CK7-positive bile duct epithelial cells, whereas this phenomenon was not observed in the livers derived from mice implanted with WB$^{Ctrl}$ cells (Supplementary Fig. 3d).

To further confirm our observation, we performed an intrasplenic injection of WB$^{Ctrl}$/WB$^{BMI1}$ cells into nude mice (Fig. 3f), indicating that ~17% of the mice injected with WB$^{BMI1}$, but not the mice injected with WB$^{Ctrl}$ cells, developed jaundice, and had an elevated bilirubin serum level (including both direct and total bilirubin level) 4 weeks after the injection (Fig. 3g and Supplementary Table 4). Importantly, our data revealed that intrasplenic implantation of the WB$^{BMI1}$ cells formed primary splenic tumors at 100% incidence (Fig. 3h), while 67% of the mice formed liver metastasis, and ~17% of the WB$^{BMI1}$ implanted mice had enlarged gall bladders and bile ducts (Fig. 3h). Again, histopathological study of the tumor serial sections revealed the presence of BDTT in the WB$^{BMI1}$ implanted mouse with jaundice only, while it also showed that the tumor thrombus originated from the invasion of WB$^{BMI1}$ cells from liver metastatic stie into the lumen of bile duct (Fig. 3i). This observation was supported by the subsequent IHC analysis of CK7 and CK19 expression in these tissue sections (Fig. 3i). In contrast, none of the WB$^{Ctrl}$ implanted mice developed primary splenic tumor and liver metastasis as well as BDTT (Fig. 3h, i). Overall, our data indicated that orthotopic liver/splenic implantation of BMI1$^{high}$ TICs could generate primary tumors/liver metastases and induce spontaneous BDTT. More importantly, we have established a preclinical animal model of BDTT in the field of HCC, providing an opportunity for exploring the underlying molecular mechanism of BDTT.

## Comparative proteomics analysis reveals an enrichment in CTSB mediated cellular components and pathways in BMI1$^{high}$ TICs

We next attempted to examine the molecular mechanism underlying BMI1$^{high}$ TIC mediated BDTT. By performing a comparative proteomics analysis with WB$^{Ctrl}$ and WB$^{BMI1}$ cells, we showed that the expression of a secreted factor, lysosomal cathepsin B (CTSB), was upregulated in WB$^{BMI1}$ cells as compared to WB$^{Ctrl}$ cells, while further GO (gene ontology) enrichment analysis indicated an enrichment in CTSB related cellular components and pathways, such as extracellular region, cell adhesion and cell junction, in WB$^{BMI1}$ cells (Fig. 4a, b). Lysosomal CTSB was known to be secreted by cancer cells, which could degrade extracellular matrix as well as breaking down cell adhesion and junction to stimulate cancer cell invasiveness[24,37]. Western blot analysis demonstrated a positive correlation between CTSB and BMI1

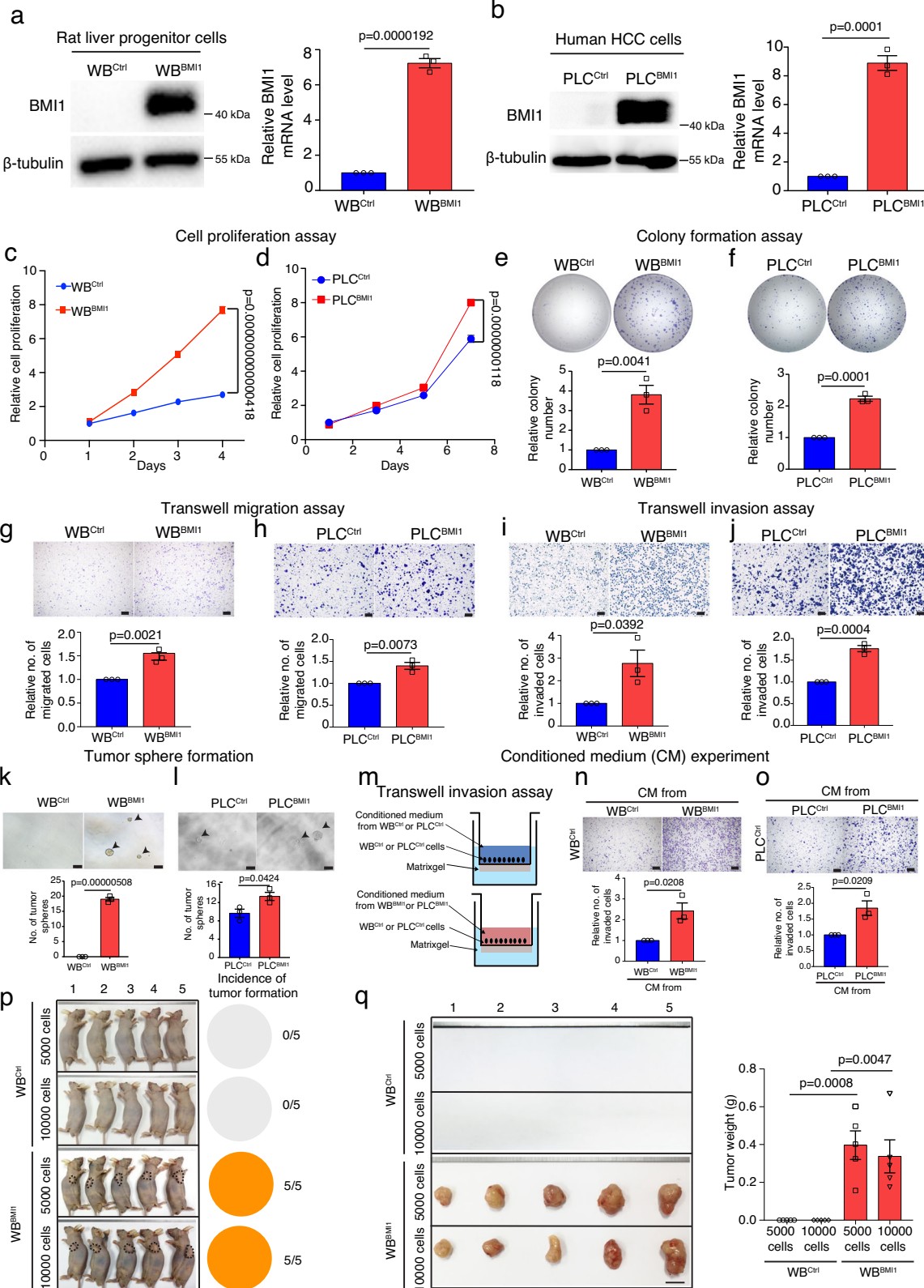

expression in a panel of HCC cell lines (Supplementary Fig. 4a), while its expression was up-regulated at the translational level, but not at the mRNA level, in WB$^{BMI1}$/PLC$^{BMI1}$ cells as compared to their control cells (Fig. 4c, d). Additionally, depleting BMI1 by shRNAs or inhibitors expression reduced CTSB expression in MHCC97H cells at the protein level only (Supplementary Fig. 4b–e). Furthermore, both ELISA and CTSB activity measurement experiments illustrated that the

expression level and enzymatic activity of CTSB was enhanced dramatically in CM harvested from WB$^{BMI1}$ or PLC$^{BMI1}$ cells when compared with CM from WB$^{Ctrl}$ or PLC$^{Ctrl}$ cells (Fig. 4e–h). In contrast, CM from BMI1 silenced MHCC97H cells or MHCC97H cells treated with BMI1 inhibitors had reduced CTSB expression and enzymatic activity compared to the CM from scramble transfected cells or untreated cells (Supplementary Fig. 4f–i). Clinically, our data demonstrated that in

**Fig. 2 | Overexpression of BMI1 spontaneously transforms liver progenitor cells into tumor initiating cells. a, b** Western blot and RT-PCR analysis of BMI1 expression in WBF344/PLC cells stably transfected with either empty vector (WB^Ctrl^/PLC^Ctrl^) or BMI1 overexpression plasmid (WB^BMI1^/PLC^BMI1^). Molecular weight markers of the ladders used (in kDa) are given on the right-hand side. Bar chart shows the relative expression of BMI1 in WB^BMI1^ cells after normalization to WB^Ctrl^ cells (*n* = 3 independent experiments). **c, d** CCK8 proliferation assay of either WB^Ctrl^/WB^BMI1^ cells or PLC ^Ctrl^/PLC^BMI1^ cells (*n* = 4 independent experiments). **e, f** Colony formation assay of either WB^Ctrl^/WB^BMI1^ cells or PLC ^Ctrl^/PLC^BMI1^ cells (*n* = 3 independent experiments). **g, h** Transwell migration assays of either WB^Ctrl^/WB^BMI1^ cells or PLC ^Ctrl^/PLC^BMI1^ cells (*n* = 3 independent experiments). **i, j** Bar charts show the relative number of invaded cells in each group (*n* = 3 independent experiments). **k, l** Bar charts show the number of tumor spheres in each group (*n* = 3 independent experiments). **m** Schematic diagram of conditioned medium experiment in conjunction with transwell invasion assays. **n, o** Bar charts show the relative number of invaded cells in each group (*n* = 3 independent experiments). **p** Nude mice were subcutaneously injected with either 5000/10,000 WB^Ctrl^ or WB^BMI1^ cells. Pie charts show the probability of tumor formation in each group. Representative gross mouse image from each group is given. Dot lines indicate the positions of subcutaneous tumors (*n* = 5 mice per group). **q** Bar charts represent the mean of tumor weight in each group (*n* = 5 mice per group). Means ± S.E.M. **a, b, e–o** Unpaired two-tailed *t* test. **q** One-way ANOVA. **c, d** Two-way ANOVA. Scale bars in (**g–o**) represent 100 μm, (**q**) 1 cm. Source data are provided in a Source data file.

HCC patients, elevated CTSB expression correlated with worse overall survival and increased cancer progression (Fig. 4i, k and Supplementary Fig. 4j), but its expression was not correlated with tumor size (Fig. 4l). Interestingly, we also showed that in HCC, high tumor expression level of CTSB was correlated with increased incidence of BDTT (Fig. 4m, n), whilst HCC patients with high CTSB expression and BDTT exhibited a relatively strong correlation with poor overall survival (Supplementary Fig. 4k). Importantly, the serum level of CTSB was up-regulated in HCC patients having BDTT (Fig. 4o). Furthermore, we showed that in HCC patients, elevated tumor expression of BMI1 and CTSB strongly associated with poor overall survival (Fig. 4p), while their expression was positively correlated too (Fig. 4q). Finally, our data showed that in HCC patients, elevated expression of TIC-related protein signature and CTSB associated strongly with worse overall survival (Fig. 4r), whilst their expression was positively correlated (Fig. 4s). To summarize, we have shown that CTSB may act as a downstream effector in BMI1-mediated HCC progression, offering the prospect of a valuable serum marker for HCC patients with BDTT.

## CTSB inhibitor treatment prohibits BDTT and extends mouse survival without affecting primary tumor growth

To determine the role of CTSB in BMI1 mediated cancer cell properties, we genetically modified CTSB expression in WB^BMI1^/PLC^BMI1^ and BMI1 depleted MHCC97H cells (Supplementary Fig. 5a–c), indicating that neither CTSB depletion/overexpression affected the proliferation and colony formation in these cells (Supplementary Fig. 5d–i). In contrast, depletion of CTSB by siRNA or an CTSB inhibitor CA-074 reduced the invasion ability of WB^BMI1^ and PLC^BMI1^ cells as compared to non-silencing control siRNA (siNSC) transfected/placebo treated cells (Fig. 5a–d), while overexpression of CTSB in BMI1 depleted MHCC97H cells enhanced cancer cell invasion when compared with control cells (Supplementary Fig. 5j). In addition, exposure of WB^Ctrl^/PLC^Ctrl^ cells with CM harvested from CTSB depleted WB^BMI1^/PLC^BMI1^ cells reduced their invasiveness as compared with the cells treated with the CM of siNSC transfected WB^BMI1^/PLC^BMI1^ cells (Fig. 5e, f). We next created an in vitro trans-intrahepatic biliary epithelial migration model in which HCC cells were placed onto a monolayer of human intrahepatic biliary epithelial cells in the upper compartment of a transwell inset coated with Matrigel matrix with/without CTSB inhibitor, and the lower compartment was loaded with DMEM plus 20% fetal bovine serum (FBS) (Fig. 5g). Strikingly, our data showed that overexpression of BMI1 enhanced the trans-intrahepatic biliary epithelial migration in human PLC cells, while depleting CTSB by siRNA or an inhibitor reduced the enhanced trans-intrahepatic biliary epithelial migration ability observed in PLC^BMI1^ cells (Fig. 5h, i). Additionally, depleting BMI1 in MHCC97H cells reduced their trans-intrahepatic biliary epithelial migration ability in comparison with scramble transfected cells (Fig. 5j), while overexpression of CTSB in BMI1 depleted MHCC97H displayed an opposite effect (Supplementary Fig. 5k).

To further examine our observation in vivo, we subcutaneously injected WB^BMI1^ cells into nude mice which were then treated with placebo or CTSB inhibitor CA-074. As expected, CA-074 treatment showed no effect on the growth of WB^BMI1^ subcutaneous tumors (Fig. 5k, l). In contrast, administration of CA-074 reduced the incidence of either jaundice, elevated bilirubin level, gallbladder and bile duct enlargement or BDTT in WB^BMI1^ orthotopic tumor bearing mice in comparison with placebo treated mice (Fig. 5m–o and Supplementary Table 5), while it did not affect tumor formation and primary tumor growth in these mice (Fig. 5n–p). Finally, we showed that CTSB inhibitor treatment extended the survival of WB^BMI1^ orthotopic tumor bearing mice compared to placebo treated group (Fig. 5q). Overall, these findings indicate that CTSB plays an important role in BMI1 dependent BDTT, identifying a potential therapeutic target for HCC patients with BDTT.

## BMI1^high^ TICs repress miR-218-1-3p expression to promote CTSB driven bile duct tumor thrombosis

In the current study, our data demonstrated that BMI1, a regulator of miRNA expression[38], modulated CTSB expression at the protein level. We therefore used CTSB mRNA sequence to predict its regulatory miRNAs using three different online search tools, while Venn diagram analysis indicated that 41 of these predicted miRNAs were commonly identified by all these tools (Fig. 6a). We next performed RT-PCR quantification of these miRNAs in BMI1 genetically modified HCC cells. In particular, the expression of miR−218-1-3p was down-regulated in BMI1 stably expressing PLC cells in relation to control cells, whilst its expression was dramatically increased in BMI1 depleted MHCC97H cells (Fig. 6b). Venn diagram analysis indicated that miR-218-1-3p was the only miRNA expression of which was correlated with BMI1 expression (Fig. 6c). We next analysed the promoter of miR-218-1-3p using the PRE sequences obtained from a previous publication[16], indicating that its promoter contained 2 putative PRE sites (Fig. 6d), while chromatin immunoprecipitation (ChIP) assays indicated the increased binding of BMI1 and trimethylated H3K27 on these two PRE sites in PLC^BMI1^ cells relative to PLC^Ctrl^ cells (Fig. 6e). Furthermore, western blot analysis revealed an increase in trimethylated H3K27 levels in BMI1 overexpressing PLC cells (PLC^BMI1^) compared to empty vector-transfected cells (PLC^Ctrl^) (Supplementary Fig. 6a). These results suggested that BMI1 may collaborate with histone methyltransferase EZH2 to repress miR-213-1-3p expression by remodeling chromatin structure. To substantiate this discovery, we conducted a series of biochemical analyses using EZH2 inhibitor PF-06726304. Our results demonstrated that EZH2 inhibition led to a reduction in trimethylated H3K27 levels in BMI1 overexpressing PLC cells relative to placebo treated cells (Supplementary Fig. 6b). Additionally, it resulted in decreased binding of trimethylated H3K27 to the two PRE sites within the miR-218-1-3p promoter (Supplementary Fig. 6c). Subsequent RT-PCR analysis indicated an up-regulation of miR-218-1-3p in EZH2 inhibitor treated BMI1 overexpressing PLC cells and MHCC97H cells compared to placebo treated cells (Supplementary Fig. 6d). To investigate the regulatory role of miR-218-1-3p on CTSB expression, we overexpressed it in WB^BMI1^, MHCC97H and PLC^BMI1^ cells respectively (Supplementary Fig. 6e), showing that the expression of CTSB was reduced in miR-218-1-3p mimic expressing WB^BMI1^/MHCC97H cells/

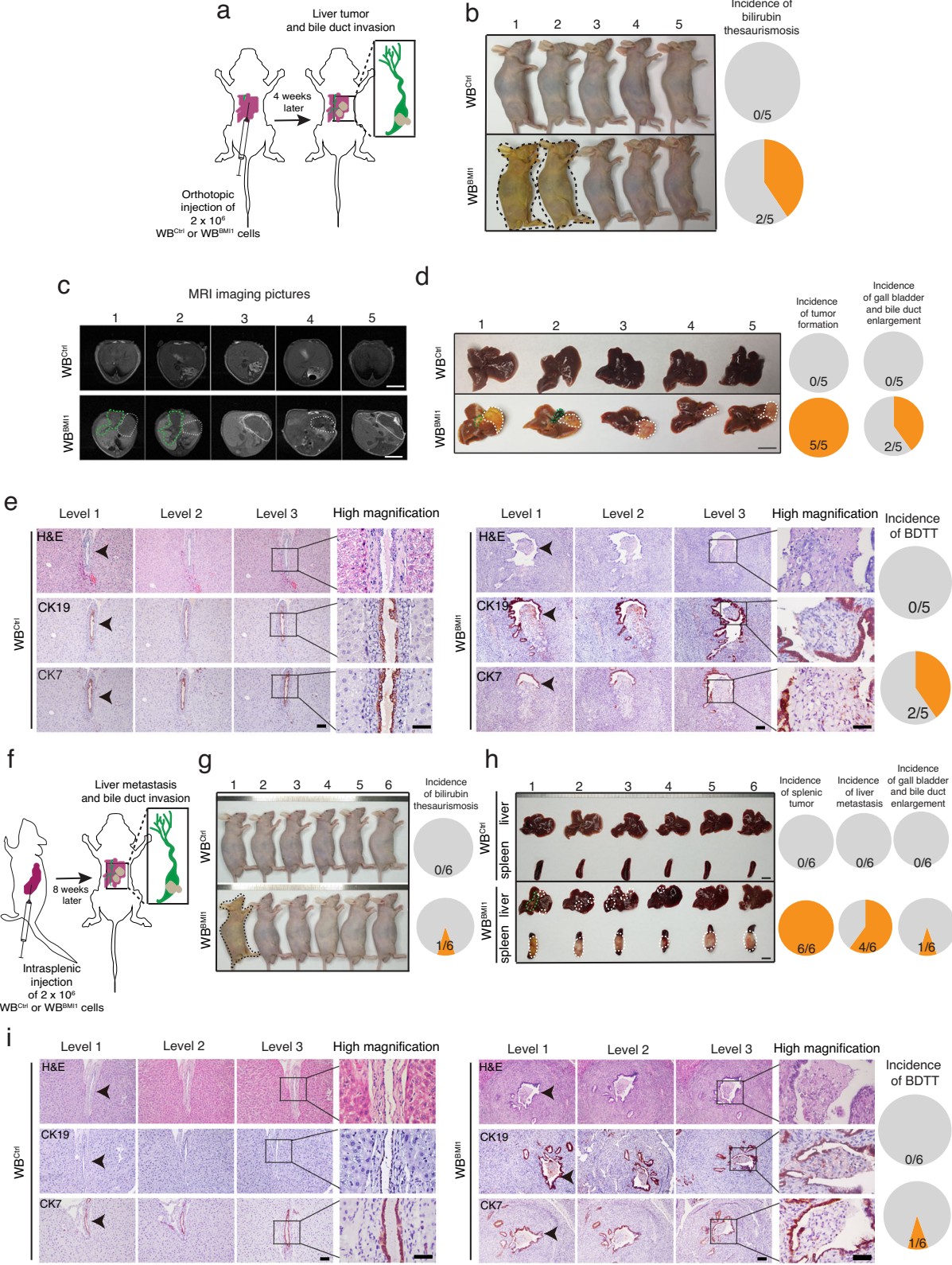

PLC^BMI1 cells relative to negative control (NC) mimic transfected cells (Fig. 6f, g and Supplementary Fig. 6f, g). We then exploited the potential downstream targets of human miR-218-1-3p by utilizing the TargetScan search tool[39], indicating that lysosomal CTSB was one of the potential targets due to its 3'UTR region of CTSB carrying a conserved miR-218-1-3p seed-matching sequence (Fig. 6h). To determine whether miR-218-1-3p bound with the 3'UTR binding sites of CTSB, we

co-transfected MHCC97H and PLC^BMI1 cells with a luciferase reporter vector construct containing an insert of either WT or mutated CTSB 3'UTR binding site together with miR-218-1-3p mimics or NC mimics. Strikingly, overexpression of miR-218-1-3p repressed the luciferase activity in MHCC97H and PLC^BMI1 cells that transfected with the reporter vector carrying WT-CTSB 3'UTR binding sequence, while it did not influence the luciferase activity in the cells transfected with the

**Fig. 3 | Orthotopic implantation of WB^BMI1 cells into the livers/spleens of mice can form primary liver tumors/metastases and induce spontaneous bile duct tumor thrombosis. a** Schematic diagram of the procedure for orthotopic injection of WB^Ctrl or WB^BMI1 cells into the Glisson's capsule of liver in a nude mouse. **b** Representative picture of the mice implanted with WB^Ctrl or WB^BMI1 cells 28 days post-injection. Black dotted lines indicate the mice with bilirubin thesaurismosis (jaundice). Pie charts indicate the incidence of jaundice from each group (n = 5 mice per group). **c** Representative MRI pictures of the mice implanted with WB^Ctrl or WB^BMI1 cells 4-week post-injection are given. Green dotted lines indicate gallbladder and bile duct enlargement and white dotted lines for tumor positions (n = 5 mice per group). **d** Representative gross images of livers derived from each group are given. Pie charts indicate the incidence of liver tumor formation and gallbladder and bile duct enlargement in each group (n = 5 mice per group). **e** Representative pictures of H&E or CK7 and CK19 immunohistochemical staining in serial tissue sections derived from the livers of mice injected with either WB^Ctrl or WB^BMI1 cells are given. Arrow indicates the position of bile duct. Pie charts indicate the incidence of BDTT in each group (n = 5 mice per group). **f** Schematic diagram of the procedure of intrasplenic injection in a nude mouse. **g** Representative pictures of the mice after intrasplenically injected with WB^Ctrl or WB^BMI1 cells are given. Pie charts indicate the incidence of jaundice in each group (n = 6 mice per group). **h** Pie charts indicate the incidence of splenic tumor formation, liver metastasis and gallbladder and bile duct enlargement in each group (n = 6 mice per group). **i** Representative pictures of H&E or CK7/CK19 IHC staining in serial tissue sections from indicated groups are given. Pie charts indicate the incidence of BDTT in each group (n = 6 mice per group). Scale bars in (**c, d, h**) represents 1 cm, (**e, i**) 100 μm. Source data are provided in a Source data file.

reporter vector construct containing mutated CTSB 3'UTR binding site sequence (Fig. 6i). Moreover, our findings demonstrated that treatment with an EZH2 inhibitor repressed luciferase activity in MHCC97H and PLC^BMI1 cells transfected with the reporter vector containing the wild-type CTSB 3'UTR binding sequence (Supplementary Fig. 6h). In contrast, it had no discernible effect on luciferase activity in cells transfected with the reporter vector construct harboring the mutated CTSB 3'UTR binding site sequence (Supplementary Fig. 6h). These results suggest a collaborative regulation of CTSB expression by BMI1 and EZH2 through miR-218-1-3p.

We next attempted to investigate the functional role of miR-218-1-3p in BMI1 mediated tumorigenesis, showing that aberrant expression of miR-218-1-3p had no effect on the cell proliferation and colony formation in WB^BMI1, MHCC97H and PLC^BMI1 cells (Supplementary Fig. 6i, j), whereas it decreased the invasion ability in WB^BMI1, MHCC97H and PLC^BMI1 cells (Supplementary Fig. 6k). Moreover, we showed that exposure of WB^Ctrl and PLC^Ctrl cells with CM from miR-218-1-3p mimic expressing WB^BMI1 and PLC^BMI1 cells reduced their invasion ability as compared to the cells treated with CM from NC mimic transfected WB^BMI1 and PLC^BMI1 cells respectively (Supplementary Fig. 6l). Lastly, miR-218-1-3p overexpression prohibited the trans-intrahepatic biliary epithelial migration in PLC^BMI1 cells in comparison with NC mimic transfected cells (Supplementary Fig. 6m). By performing a florescence in-situ hybridization (FISH) experiment of miR-218-1-3p with HCC tumor sections, we showed that in HCC tumors, the expression level of miR-218-1-3p was correlated negatively with BMI1/CTSB expression respectively (Fig. 6j).

To examine the importance of miR-218-1-3p expression in BDTT in vivo, we orthotopically injected miR-218-1-3p or NC mimics stably transfected WB^BMI1 cells into the livers of nude mice, indicating that miR-219-1-3p overexpression reduced the incidence of either jaundice, elevated bilirubin, gall bladder and bile duct enlargement or BDTT without affecting primary liver tumor growth in WB^BMI1 implanted mice as compared with NC mimic transfected group (Fig. 6k–n). Combined histopathological and immunohistochemical staining analysis of the serial tumor sections indicated that miR-218-1-3p overexpression prohibited BDTT in the WB^BMI1 orthotopic liver tumor bearing mice (Supplementary Fig. 6n), while this observation was further confirmed by the results of their liver function blood test (Supplementary Table 6).

Overall, our works revealed that BMI1 overexpression malignantly transformed LPCs into BMI1^high TICs, while orthotopic liver implantation of these cells in mice could generate primary tumors and spontaneous BDTT. Mechanistically, BMI1 silenced miR-218-1-3p expression via PRC dependent mechanism, which then epigenetically upregulated CTSB secretion in BMI1^high TICs to facilitate their invasion into bile ducts to form BDTT (Fig. 7).

## Discussion

We report here that the development of BDTT in HCC patients worsens disease prognosis in HCC patients. This occurrence correlates with the expression of BMI1 and TIC-related gene/protein signature. We have identified the presence of BMI1^high TICs in HCC patients with BDTT, suggesting their role in initiating the pathogenesis of BDTT in these patients. Our findings reveal the molecular mechanism underlying BDTT in HCC biology and may enable the identification of these HCC patients using a potential serum marker CTSB. Overall, our data open new routes for the diagnosis, classification, and personalized treatment strategy of HCC patients with BDTT.

BDTT is one of the salient features in HCC patents, which is hardly diagnosed and treated with conventional therapies[8,9,40]. Obstructive jaundice is the predominant clinical presentation of HCC patients with BDTT, whereas it can be easily misdiagnosed as bile duct cancer or biliary stones[4,5]. There has been no improvement in the diagnosis and treatment of this disease over the past decade. Although several studies propose that TICs derived from malignantly transformed LPCs might contribute to BDTT[10,12], their roles are unclear due to lack of a preclinical HCC animal model accompanying with BDTT. Interestingly, our previous research demonstrated that dysregulation of BMI1 expression could induce malignant transformation of rat LPCs[12]. Consistently, Chen et al. show that BMI1 overexpression promotes cancer stemness in squamous cell carcinoma[14], while its expression also regulates the migration and invasion of TICs in colon cancer[11]. Nevertheless, none of these studies have examined the role of BMI1^high TICs in hepatocarcinogenesis and BDTT.

In this research work, we provide strong clinical data showing that in HCC, high expression level of BMI1 and/or TIC-related gene/protein signature associates with worse prognosis and increased BDTT incidence in a comparatively large patient cohort, implying the possible role of BMI1^high TICs on HCC development and progression. We and others have shown that BMI1 expression can regulate cancer cell proliferation, migration, and invasion[12], whereas its regulatory role in TIC properties and BDTT is less clear. We find that BMI1 overexpression malignantly transforms LPCs into BMI1^high TICs, while these cells display strong tumorigenicity and increase their trans-intrahepatic biliary epithelial migration ability through paracrine effect. We show that orthotropic implantation of WB^BMI1 cells into the livers or spleen of nude mice can generate liver tumors/metastases and BDTT. We also present histopathological and immunohistochemical/co-immunostaining evidence, along with cell tracing experiments, demonstrating the invasive growth of BMI1^high TICs from the primary liver tumor or metastatic site into the bile duct lumen, leading to BDTT. Furthermore, we have successfully developed HCC mouse models featuring BDTT in the existing literature.

We then sought to delve into the molecular mechanism behind BMI1^high TIC-driven BDTT by conducting proteomics and biochemical analyses. These investigations revealed an up-regulation in the expression and enzymatic activity of lysosomal CTSB in BMI1^high TICs. Interestingly, recent studies have revealed that colon cancer cells can secrete CTSB to degrade extracellular matrix and break down cell junctions, thereby enhancing cell motility[26]. Furthermore, inhibition of CTSB has been demonstrated to reduce the invasiveness of these cells

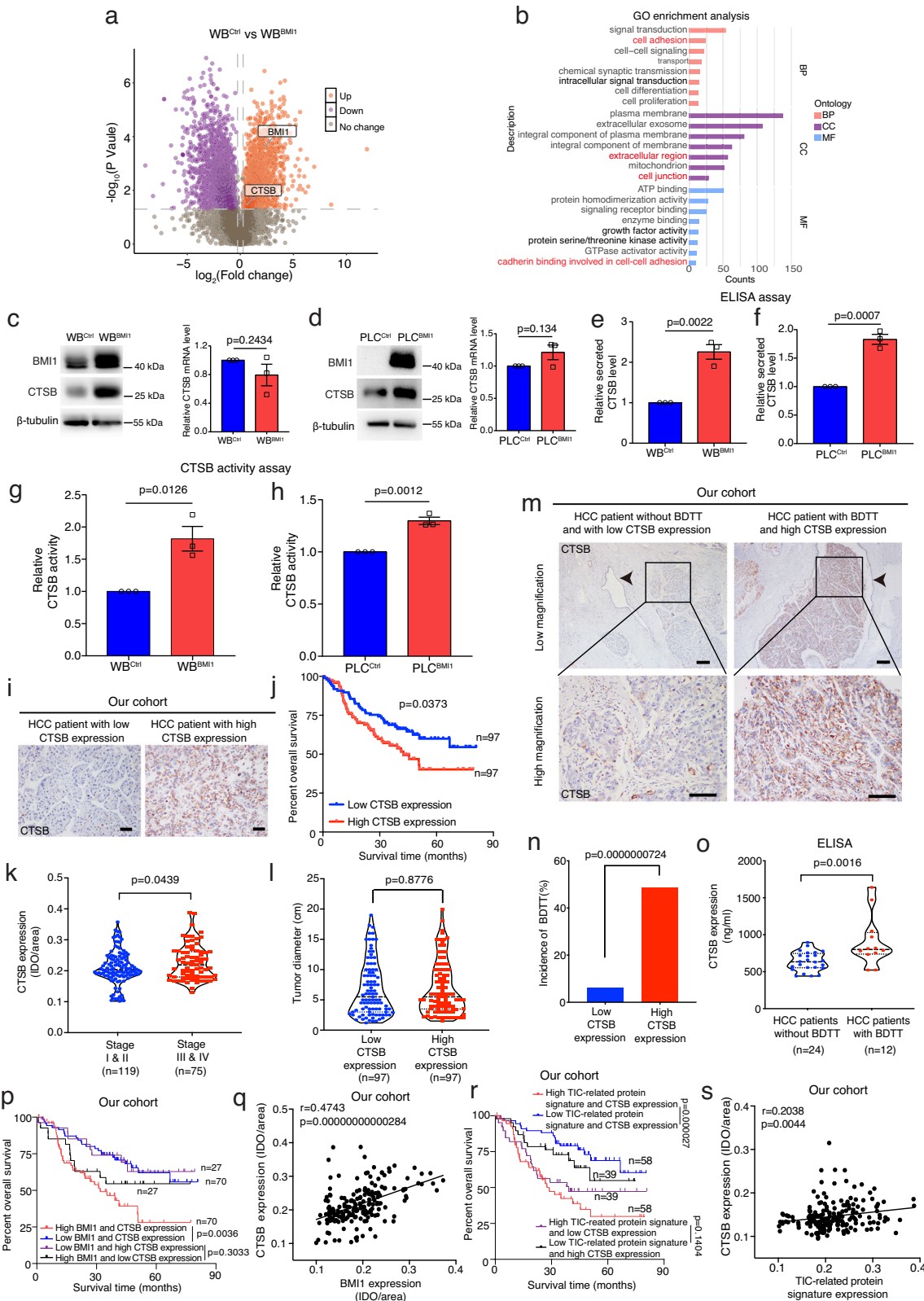

in vitro[37]. Notably, CTSB can also facilitate cancer cell invasion by degrading the basement membrane[28]. Coincidently, we show that depletion of CTSB reduces the invasion and trans-intrahepatic biliary epithelial migration in BMI1 expressing cells, while it has no effect on their proliferation and colony formation. Furthermore, our data show that treatment with CM from CTSB depleted BMI1 expressing cells decreases the trans-intrahepatic biliary epithelial migration and

invasion in empty vector transfected cells as compared with the cells exposed to CM from siNSC transfected BMI1 expressing cells. In vivo, we also show that CTSB inhibitor treatment prohibits BDTT and prolongs mouse survival without affecting primary orthotopic liver tumor growth. It is therefore tempting to suggest that BMI1^high TICs secrete CTSB to break down bile duct epithelium to facilitate their bile duct invasion. Clinically, we show that high CTSB expression associates with

**Fig. 4 | Comparative proteomics analysis reveals an enrichment in CTSB dependent cellular components and pathways in BMI1[high] TICs. a, b** Volcano plot and GO enrichment analysis of the proteomics data from WB[Ctrl] and WB[BMI1] cells (*n* = 3 independent experiments). Red color highlights CTSB related cellular components and pathways. BP stands for biological process, CC for cellular component, MF for molecular function. **c, d** Western blot and RT-PCR analysis of CTSB expression in either WB[Ctrl] and WB[BMI1] cells or PLC[Ctrl] and PLC[BMI1] cells (*n* = 3 independent experiments). Bar charts represent means ± S.E.M. **e–h** CTSB ELISA and activity measurement analysis of the conditioned medium derived from each group (*n* = 3 independent experiments). Bar charts represent means ± S.E.M. **i–l** Correlation between CTSB expression and overall survival/cancer progression/tumor size in HCC patients (*n* = 194 HCC patients, our cohort). **m, n** Correlation between CTSB expression and BDTT incidence in HCC patients. Representative IHC pictures of CTSB staining in tumor sections derived from each group are given. Arrow indicates the position of bile duct. **o** ELISA assay analysis of the serum CTSB level in HCC patients with/without BDTT (*n* = 36 HCC patients, our cohort). **p** High BMI1 and CTSB expression strongly correlated with poor overall survival in HCC patients as compared to patients with low BMI1 and CTSB expression (*n* = 194 HCC patients, our cohort). **q** Correlation study between BMI1 and CTSB expression in HCC patients from our cohort (*n* = 194 HCC patients). **r** High TIC-related protein signature and CTSB expression associated with poor overall survival in HCC patients when compared to patients with low TIC-related protein signature and CTSB expression (*n* = 194 HCC patients, our cohort). **s** Correlation between TIC-related protein signature and CTSB expression in HCC patients (*n* = 194 HCC patients). **c–h, k, l, o** Unpaired two-tailed *t* test. **j, p, r** Log-rank (Mantel-Cox) test. **n** Two-sided Chi-square test. **q, s** Two-sided Pearson correlation test. Scale bars in (**i, m**) represents 100 μm. Source data are provided in a Source data file.

increased incidence of BDTT in HCC patients, while the HCC patients with BDTT have an elevated serum CTSB level, suggesting that CTSB could potentially function as a diagnostic marker and therapeutic target for HCC patients with BDTT and warrants further investigation.

Since BMI1 can collaborate with histone methyltransferase EZH2 to regulate miRNA expression via PRC dependent effect[17,20], and together with the fact that BMI1 modulates CTSB expression translationally, there is a possibility that miRNA involves in its regulatory role on CTSB expression. Indeed, CTSB expression has been predicted to be regulated by miRNAs such as miR-218 family members in medulloblastoma[29]. Our ChIP assays uncovered elevated occupancy of BMI1 and trimethylated H3K27 on the promoter's PRE sites of miR-218-1-3p in BMI1-expressing cells. However, treatment with EZH2 inhibitors diminished the augmented binding of H3K27me3 to these PRE sites seen in BMI1-expressing cells. Importantly, our data illustrate that overexpression of miR-218-1-3p inhibits CTSB secretion in BMI1 expressing cells, while its expression also affects their invasion and trans-intrahepatic biliary epithelial migration ability. In support of this finding, Guo et al. shows that miR-218-1-3p regulates lung cancer cell invasiveness[41], while its expression has been linked with epithelial mesenchymal transition and angiogenesis in colorectal cancer[42]. We also discover that miR-218-1-3p overexpression reduces jaundice, gall bladder and bile duct enlargement and BDTT in WB[BMI1] implanted mice, independent of primary orthotopic liver tumor growth. Clinically, the tumor expression between miR-218-1-3p and BMI1/CTSB is negatively correlated in HCC patients. These data imply that BMI1 up-regulates CTSB secretion in TICs by repressing miR-218-1-3p expression, indicating that epigenetic regulation of CTSB secretion in BMI1[high] TICs determines BDTT.

In summary, our findings solve the unsolved puzzle about the cellular origin and pathogenesis of BDTT in the field of HCC biology, indicating that BMI1[high] TICs induce hepatocarcinogenesis and undergo CTSB driven bile duct invasion to form tumor thrombus. More importantly, our work offers a potential diagnostic marker and treatment avenue for HCC patients with BDTT, opening the door for future clinical applications.

## Methods

### Ethical statement
All animal procedures carried out in this study were approved by the institutional animal care and use committee of Sun Yat-sen university (SYSU-IACUC-2021-B0173). All HCC samples were collected with written informed consent and approved by the ethnical review committee of Sun Yat-sen Memorial Hospital in Guangzhou, China (SYSKY-2023-489-01).

### Clinical specimen collection and online database analysis
For our own patient cohort study, it involves HCC patients who underwent surgical resection at Sun Yat-sen Memorial Hospital of Sun Yat-sen University between 2012 and 2018. During this period, a total of 1791 cases of HCC patients underwent liver cancer resection surgery. Among them, there were 53 cases of patients with combined bile duct cancer thrombi. The inclusion criteria for this study were: 1. Pathological diagnosis of liver cancer; 2. R0 resection; 3. Complete clinical data and follow-up information; 4. Specimens available. Exclusion criteria included: 1. Preoperative radiotherapy or chemotherapy; 2. Concurrent with other tumors. Among them, the diagnostic criteria for HCC patients with combined bile duct cancer thrombi were: observation of tumor foci within the bile duct lumen under the microscope, with pathological diagnosis of liver cancer combined with bile duct cancer thrombi. Based on the above selection criteria, 151 cases of patients without bile duct cancer thrombi and 43 cases of patients with combined bile duct cancer thrombi were included in this study. The study was conducted continuously with these 194 cases of patients with HCC. The clinicopathological characteristics of these patients, such as gender, number, age, alcohol consumption, presence of cirrhosis, tumorous number, tumor size, BDTT under the microscope, AFP level, vascular invasion and TNM stage, are provided in Supplementary Table 1. All HCC samples were anonymized in accordance with local ethical guidelines, as required by the Declaration of Helsinki. For online database analysis, TCGA database was utilized to obtain the RNA-seq and clinicopathological data of human patients with HCC (TCGA-LIHC)[30], which was then analyzed as previously described[17,43]. Briefly, we utilized the median cutoff function provided by the Kaplan–Meier (KM) plotter web tool (http://kmplot.com/analysis/) to categorize HCC patients based on their mean expression of the TIC gene signature (CD44/CD133/SOX9) or BMI1 and TIC gene signature expression, following the guidelines and database established by the web tool developer[31]. For other studies, the median mRNA expression level of BMI1 or CTSB was employed as thresholds to categorize patients into groups with high or low expression.

### Tissue culture
Rat LPC WB-F344 was purchased from the Cell Bank of the Chinese Academy of Science (Cat no.#CTCC-400-0377, Shanghai, China). BMI1 stably expressing WB-F344 cells (WB[BMI1]) and empty vector transfected WB-F344 cells (WB[Ctrl]) were generated and cultured as described previously[12]. Human intrahepatic biliary epithelial cells (HIBEpiC) were purchased from ScienCell (Cat no.#5100). Human HCC cell lines (i.e., PLC (Cat no.#CTCC-003-0017), Huh7 (Cat no.#CTCC-003-0019) and HepG2 (Cat no.#CTCC-001-0014), MHCC97H (Cat no.#CTCC-400-0192)) were obtained from the Cell Bank of the Chinese Academy of Sciences (Shanghai, China). Cell lines were used as provided commercially and no additional identification was performed. No mycoplasma contamination was detected throughout the study. For the culturing conditions, cells were cultured in DMEM plus 10% FBS and 1% penicillin and streptomycin, which were maintained in a 5% $CO_2$ incubator at 37 °C. HIBEpiCs were cultured in epithelial cell medium (Cat# 5100, ScienCell). For the BMI1 overexpression, human HCC cells were transfected with pCDH-CMV-GFP-Puro-BMI1 or empty

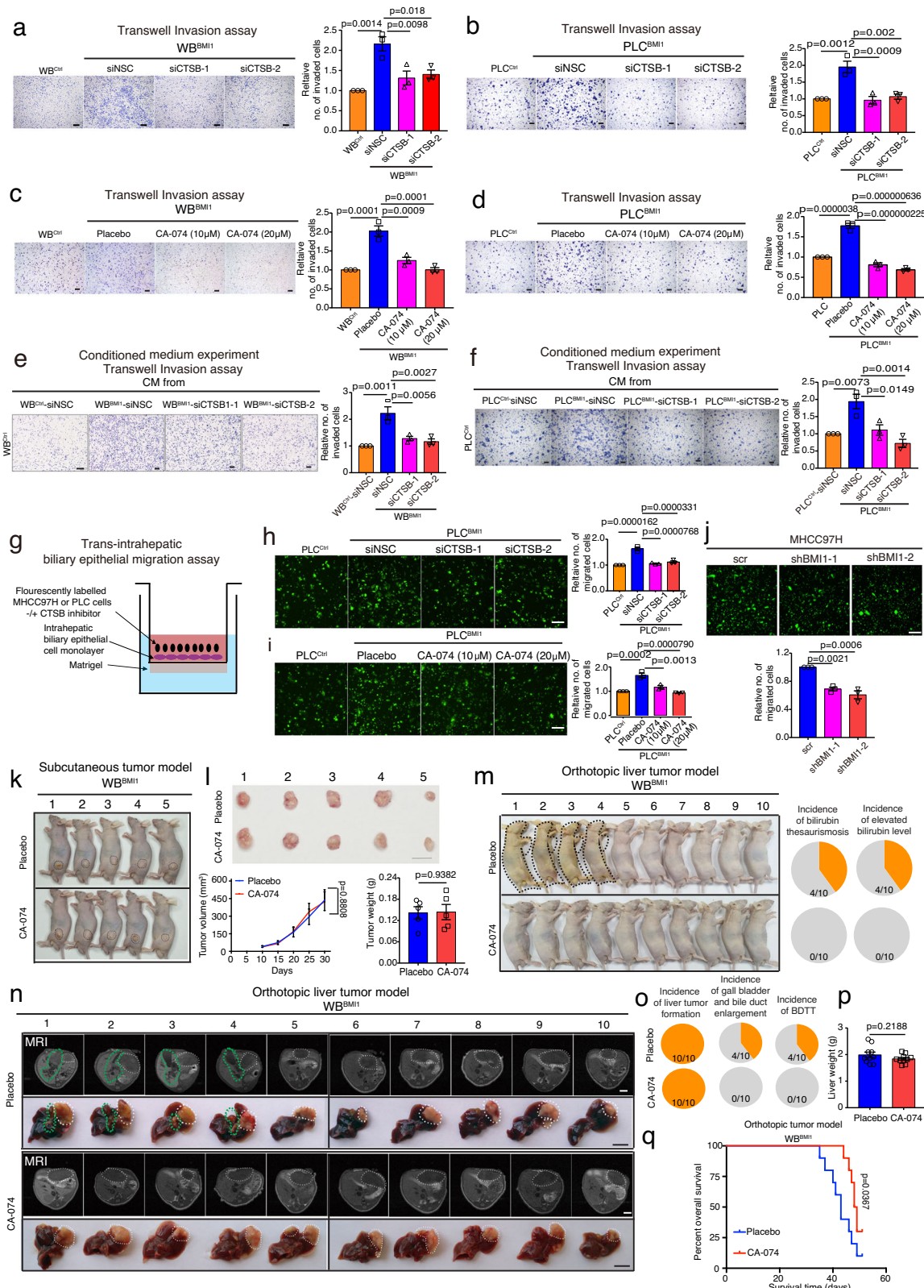

vector (Genechem, China), which were then selected with 5 µg/mL puromycin up to 2 weeks. For knocking down BMI1 expression, HCC cells were transfected with either pSuper-retro-neo-shBMI1-1/-2 (GenePharma, China) or scramble control vector, and then treated with puromycin. For the miR-218-1-3p overexpression experiment, WB^BMI1 cells were first transfected with either pCDH-CMV-GFP-Puro-miR-218-1-3p or pCDH-CMV-GFP-Puro-negative control (NC) plasmid

(GenePharma, China) and then subjected to puromycin selection. Human MHCC97H or PLC^BMI1 cells were transiently transfected with 40 nM miR-218-1-3p mimics or negative control (NC) mimics (Gene-Pharma, China) using Lipofectamine™ 3000 transfection reagent (ThermoFisher Scientific-#L3000015). For in vivo cell tracking experiments, WB^Ctrl or WB^BMI1 cells were initially seeded in six-well plates and subsequently infected with a GFP-encoding virus

**Fig. 5 | CTSB inhibitor treatment prohibits bile duct tumor thrombosis in WB^BMI1 tumor bearing mice independent of primary tumor growth.**
**a**, **b** Transwell invasion assays of WB^BMI1 or PLC^BMI1 cells stably transfected with CTSB targeting siRNA-1/-2 or non-silencing control siRNA (siNSC) (n = 3 independent experiments). **c**, **d** Transwell invasion assays of WB^BMI1 or PLC^BMI1 cells -/+ CTSB inhibitor CA-074 (n = 3 independent experiments). **e**, **f** Conditioned medium experiment. Bar charts show the relative number of invaded cells in each group (n = 3 independent experiments). **g** Schematic diagram of the trans-intrahepatic biliary epithelial migration assay. **h**–**j** Trans-intrahepatic biliary epithelial migration assays of fluorescently labeled PLC^BMI1/MHCC97H cells after either transfected with CTSB/BMI1 targeting siRNA/shRNA or non-silencing siRNA (siNSC)/scramble shRNA or treated with placebo/CTSB inhibitor at indicated doses. Representative images of fluorescently labeled migrated cells from each group are given (n = 3 independent experiments). **k**, **l** Representative pictures of tumor bearing mice and gross tumor images from each group are given. Tumor growth curves show the tumor growth in each group. Bar chart indicates the mean tumor weight from each group (n = 5 mice per group). Means ± S.E.M. **m**, **n** Representative pictures of tumor bearing mice, MRI and gross livers and spleens are given. Pie charts indicate the incidence of jaundice and elevated bilirubin level in each group (n = 10 mice per group). **o** Pie charts indicate the incidence of liver tumor formation, gall bladder and bile duct enlargement and BDTT in each group (n = 10 mice per group). **p** Bar charts indicate the mean of liver weight in each group (n = 10 mice per group). Means ± S.E.M. **q** Kaplan–Meier survival study of WB^BMI1 orthotopic liver tumor bearing mice after treated with placebo or CTSB inhibitor (n = 10 mice per group). **a**–**f**, **h**–**j** One-way ANOVA. **l** Two-way ANOVA (line graph). **l**, **p** Unpaired two-tailed t test (Bar charts). **q** Log-rank (Mantel–Cox) test. Scale bars in (**a**–**f**, **h**–**j**) represent 100 μm, (**l**, **n**) 1 cm. Source data are provided in a Source data file.

(purchased from GenePharma, China) in the presence of 6 μg/mL polybrene at 37 °C. Following a 24-h incubation period, the culture medium was replaced with fresh medium and subjected to selection with 2 μg/ml puromycin (InvivoGen-#ant-pr-1) for two weeks. The selected cells were then expanded and utilized in the specified experiments. For the CTSB knock-down experiments, cells were transiently transfected with either CTSB targeting siRNA (GenePharma, China) or non-silencing control siRNA (siNSC) using Lipofectamine™ 3000 transfection reagent (ThermoFisher Scientific-#L3000015).

### In vitro tumorigenesis assays

All experiments were carried out according to previous published methods[44]. Briefly, cell proliferation experiments were done by using Cell Counting Kit-8 (CCK8) (Cat# CK04-13, Dojindo Molecular Technologies). For the colony formation assay, single cell suspension was placed in 6-well plates at a concentration of $1.5 \times 10^3$ cells per well and left for 3-week period. After that, cell colonies were first fixed with 4% formaldehyde for 20 min and then stained with 1% crystal violet (Solarbio) for 10 min. The number of cell colonies were determined by utilizing fluorescence enzyme-linked immunospot analyzer (AID vSpot Spectrum). For the tumor sphere formation assay, a total of 800 cells were seeded onto 96-well ultra-low attachment plates, while they were grown in Dulbecco's modified Eagle medium/F12 plus 2% B27 supplement (ThermoFisher Scientific-#17504044), 20 ng/mL epidermal growth factor (PeproTech), and 20 ng/ml fibroblast growth factor (PeproTech). After 7–14 days, the sphere count was carried out under a microscope. In the study of transwell migration assay, migration activity was measured by using transwell chambers with 8 μm pore size (Corning). Briefly, 300 μL serum-free medium containing $1 \times 10^5$ cells was added to the upper compartment of transwell inserts in the presence or absence of 10/20 μM cathepsin B inhibitor CA-074 (APEXBIO-#A1926), and the bottom compartment was loaded with 700 μL complete medium with 10–20% FBS. For the transwell invasion assay, the upper compartment of a transwell insert was first coated with a layer of Matrixgel matrix (Corning-#354230). After 48 h incubation, a cotton swab was used to remove the non-migrated/-invaded cells on the upper chamber. The cells adhering to the lower chamber were fixed in the presence of 4% paraformaldehyde for 20 min, which were then counterstained with 1% crystal violet for 10 min. For the CM experiment, cells were placed onto 6-well plates at a concentration of $1.5 \times 10^6$ per well and left untouched overnight, which were then cultured in 1.5 mL serum-free medium for 24 h. The next day, the upper compartment of a transwell insert was coated with Matrigel (50 μg/mL, Corning) for 3 h. Afterwards, the cell supernatant was collected, which was then used for resuspending HCC cells. After adjusting the cell density, 300 μL of the suspension containing $1–2 \times 10^5$ cells was seeded into the upper compartment of a transwell insert, and the bottom compartment was placed with 700 μL complete medium containing 20% FBS for 48 h. For the number of migrated/invaded cell calculation, four random microscopic fields were counted in each experimental group, while the experiments were independently repeated three times. For the BMI1 inhibitor (i.e., PCT209 (Selleck-#S7539) or PRT4165 (Selleck-#S5315) experiments, the cells were treated with either DMSO, 1/10 μM PCT209 or 25/50 μM PRT4165 for 24 h and subjected to the experiments described above. For the CM involved BMI1 inhibitor, MHCC97H cells were first pre-treated with 1 μM PCT209 or 25 μM PRT4165 for 6 h and then changed with fresh medium for 24 h. The CM was then harvested for transwell invasion assay as described above.

### Western blot analysis

Cells were first washed with PBS and then lysed with RIPA lysis buffer (Beyotime-#P0013) plus protease/phosphatase inhibitor cocktail (Cwbio-#CW2200) for 1 h on ice. The protein supernatant was carefully collected and centrifuged at 12,000 g for 30 min at 4 °C, and the protein concentration was determined by using a BCA Protein Assay Kit (ThermoFisher Scientific-#23225). After heat denaturation, the equal amount of protein from each sample was resolved on 10–12% SDS-PAGE gels, which were then electroblotted onto polyvinylidene difluoride membranes (Merck Millipore-# 3010040001). The membrane was then blocked in TBS plus 0.1% Tween 20 (TBST) containing 5% milk for 1 h at room temperature and then probed with following diluted primary antibodies (1 in 1000 dilution) overnight at 4 °C: anti-BMI1 (Cell Signaling Technology-#6964), anti-Cathepsin B (Cell Signaling Technology-#31718), anti-Sox9 (Abcam-#ab185966), anti-CD44 (Cell Signaling Technology-#37259), anti-CD133 (Abcam-#ab19898), anti-β-tubulin (Abcam-#ab179513) (which were diluted in 1% (w/v) Albumin Bovine (BSA) in TBST). After washing with TBST, the membrane was incubated with horseradish peroxidase-conjugated anti-rabbit secondary antibodies (Cell Signaling Technology-#7074) for 1 h at the room temperature, which were then washed with 3X TBST. The membranes were visualized by using an enhanced chemiluminescence (ECL) system (Merck Millipore-#WBKLS0500). For the EZH2 inhibitor treatment experiment, cells were first treated with or without 15 nM PF-06726304 (Selleck-#S8494) for 24 h prior to western blot analysis. The unprocessed and uncropped scans of all western blots are provided in the Source Data File.

### RT-PCR analysis

The total RNA was isolated from cells by using RNAiso Plus reagent (Takara-#9108), 2 μg of the RNA was employed to synthesize cDNA by using a PrimeScript™ RT reagent Kit (Takara-#RR064A) and miRNA 1st strand cDNA synthesis kit (Cat, Accurate Biology-#AG11717). The quantitative RT-PCR experiments were then carried out by using SYBR green PCR master mix reagent (Takara-#639676) on a Roche Light-Cycler 480 II machine (Software: LightCycler 480 Version 1.5.1). In this study, GAPDH and U6 (small nuclear RNA) was used as an endogenous control respectively. For the EZH2 inhibitor treatment experiment, cells were first treated with or without 15 nM PF-06726304 (Selleck-#S8494) for 24 h prior to RT-PCR analysis. The sequences of RT-PCR primers utilized in this study are listed in Supplementary Table 2.

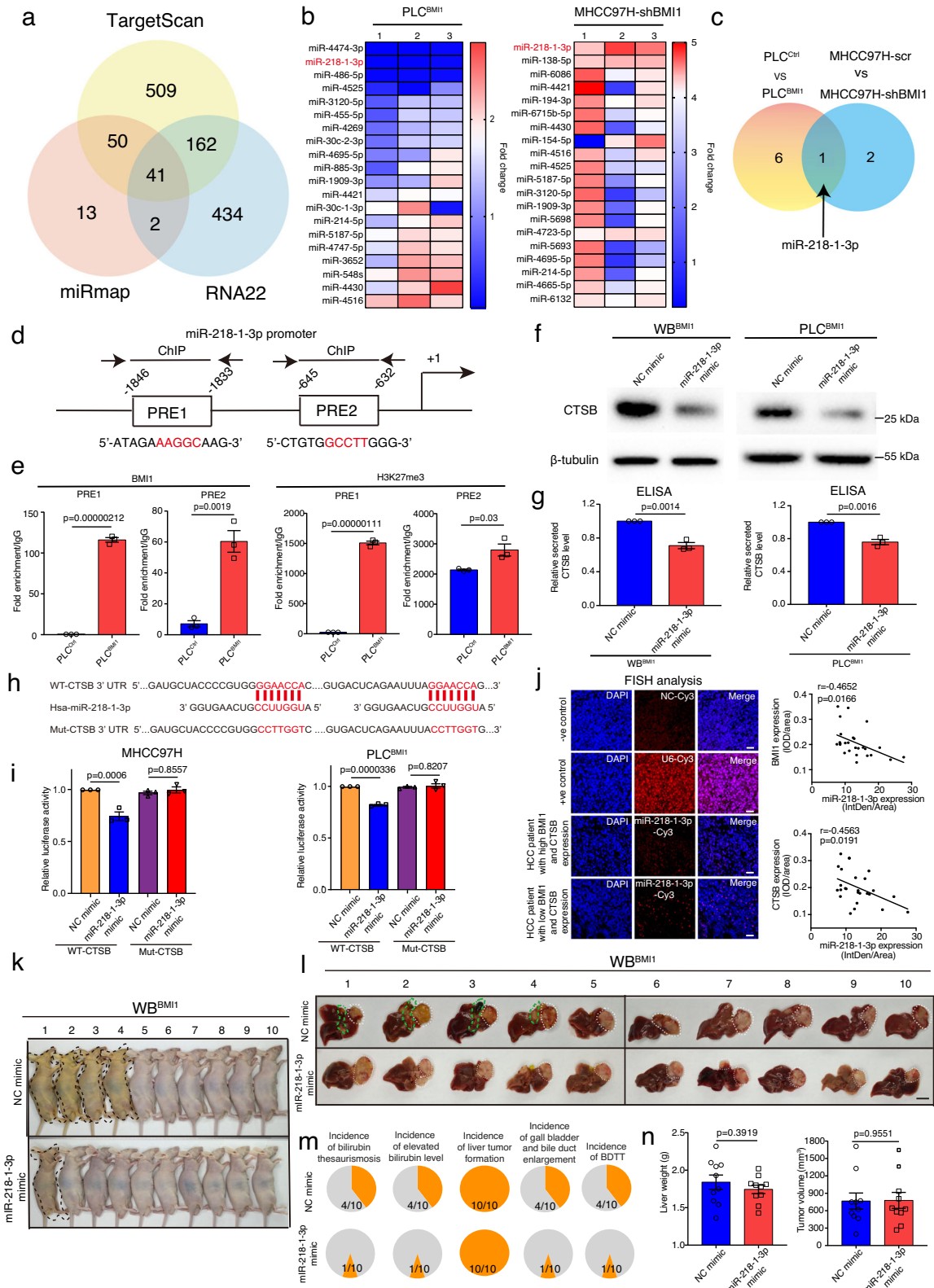

## Immunohistochemistry and Immunostaining experiments

Immunohistochemical (IHC) and immunofluorescent staining was carried out according to the standard methods. Briefly, the sections were deparaffinized, rehydrated followed by boiled in ethylenediaminetetraacetic acid (pH = 8.0) for antigen retrieval using a pressure cooker. After blocking peroxidase with hydrogen peroxide solution, the sections were incubated overnight at 4 °C with either anti-BMI1 (1:800; Cell Signaling Technology-#6964), anti-cathepsin B (1:4000; Cell Signaling Technology-#31718), anti-Sox9 (1:700; Abcam-#ab185966), anti-CD44 (1:2000; Cell Signaling Technology-#37259), anti-CD133 (1:200; Cell Signaling Technology-#64326), anti-cytokeratin 19 (CK19) (1:500; Abcam-#ab52625) and anti-cytokeratin 7 (CK7) (1:6000; Abcam-#ab181598), which were then washed and incubated with HRP-conjugated secondary antibody (DAKO) for

**Fig. 6 | BMI1 epigenetically up-regulates CTSB expression by repressing miR-218-1-3p expression. a** Venn diagram showing the number of commonly predicted CTSB targeting miRNAs from three different online search tools, including TargetScan, miRmap and RNA22. **b** Heatmaps showed the top 20 altered CTSB targeting miRNAs in PLC^BMI1 and MHCC97H-shBMI1 cells compared to PLC^Ctrl and MHCC97H-scr cells respectively (n = 3 independent experiments). **c** Venn diagram analysis revealed miR−218-1-3p as the sole miRNA correlated with BMI1 expression. **d** Schematic diagram of the predicated PRE sites in miR-218-1-3p promoter sequence. **e** ChIP assay examined BMI1 and H3K27me3 binding on miR−218-1-3p promoter's predicted PRE sites in PLC^BMI1 cells compared to PLC^Ctrl cells (n = 3 independent experiments). **f** Western blot analysis of WB^BMI1 and PLC^BMI1 cells after transfected with miR−218-1-3p mimics or NC mimics (negative control) (n = 3 independent experiments). **g** ELISA assay of CTSB levels in conditioned medium harvested from WB^BMI1 and PLC^BMI1 cells after transfected with miR−218-1-3p mimics or NC mimics (negative control) (n = 3 independent experiments). **h** Schematic diagram showing the seeding region matching between miR−218-1-3p and CTSB 3'UTR. **i** Luciferase reporter assays were performed in MHCC97H and PLC^BMI1 cells, co-transfecting them with luciferase reporter vectors containing either the wild-type (WT-CTSB) or mutated (Mut-CTSB) CTSB 3'-UTR and pLK reference plasmid, along with miR-218-1-3p mimics or NC mimics (n = 3 independent experiments). **j** Representative images of FISH analysis of miR-218-1-3p expression, negative (-ve) and positive (+ve) control probes in tumor sections from HCC patients. Correlation studies between miR-218-1-3p and BMI1/CTSB expression in HCC patient tumors (n = 26 patients, our cohort). **k**, **l** WB^BMI1 cells stably transfected with either NC mimics or miR-218-1-3p mimics were orthotopically implanted into the livers of nude mice (n = 10 mice per group). **m** Pie charts display the occurrence of the specified conditions within each group (n = 10 mice per group). **n** Bar charts show the mean of liver weight and tumor volume between groups (n = 10 mice per group). **e**, **g**, **n** Unpaired two-tailed t test. **i** One-way ANOVA test. **j** Two-sided Pearson coefficient test. Scale bars in (**j**) represent 100 μm, (**l**) 1 cm. Source data are provided in a Source data file.

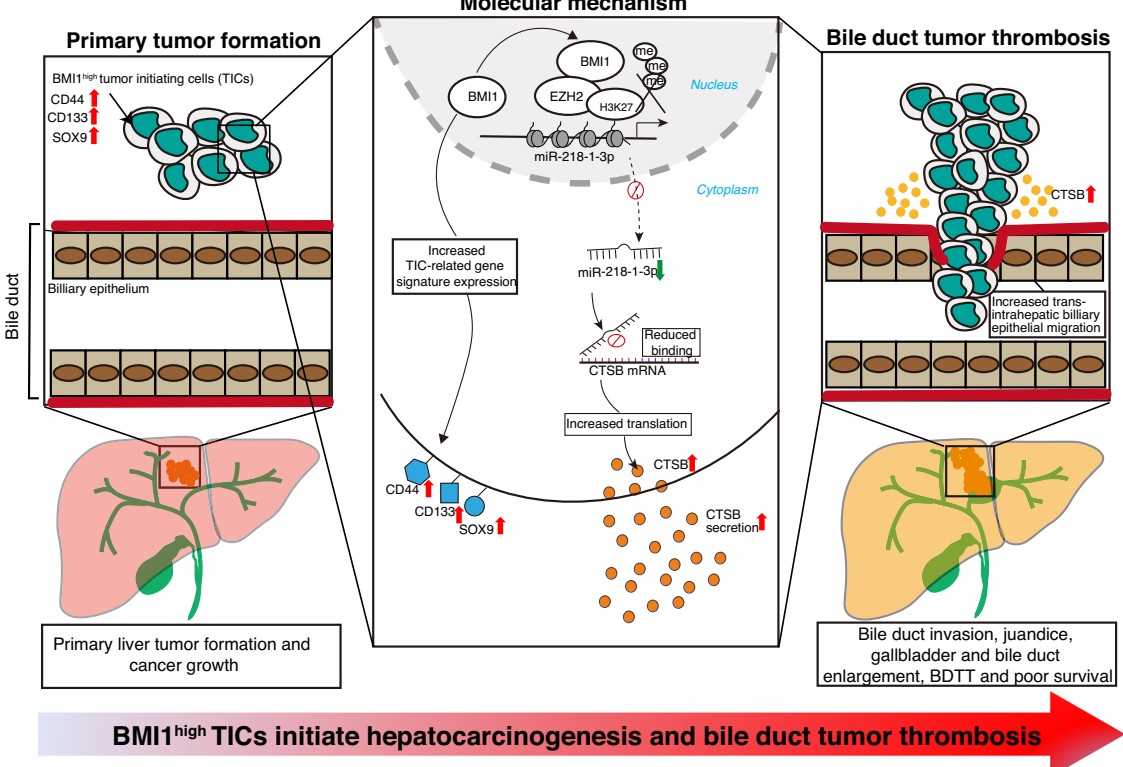

**Fig. 7 | Schematic diagram illustrates the cellular origin and molecular mechanism of bile duct tumor thrombosis in HCC.** BMI1 overexpression malignantly transformed liver progenitor cells into BMI1^high TICs, which are capable of initiating hepatocarcinogenesis and BDTT. Mechanistically, BMI1 collaborates with EZH2 to repress miR-218-1-3p expression by increasing H3K27me3 level, which in turn epigenetically up-regulates CTSB secretion in BMI1^high TICs to enhance their invasion into bile ducts to form tumor thrombi.

30 min at room temperature. Finally, the DAB staining was performed, and nuclei were counterstained with hematoxylin. The IHC or immunofluorescent images were captured using a light microscope (Microscope: Nikon NI-U; software: NIS-Elements version F 4.60.00) or a confocal microscope (Microscope: Zeiss LSM 800; software: ZEISS ZEN version 2). The intensity level of the IHC staining was quantified by using the Image-Pro Plus Version 6.0 software (IPP6, Media Cybernetics, San Diego, CA). The median of BMI1 or CTSB IHC staining was set as a cutoff threshold for stratifying patients into those with high or low expression group.

For the immunostaining experiments, tissue sections underwent deparaffinization and rehydration, followed by boiling in ethylenediaminetetraacetic acid (pH = 8.0) for antigen retrieval. After blocking peroxidase activity with hydrogen peroxide solution, the sections were incubated overnight at 4 °C with either anti-BMI1 (1:800; Cell Signaling Technology-#6964) along with anti-CD44/-CD133/-SOX9 or anti-GFP (1:1000; Cell Signaling Technology-#2555) along with anti-CK7. Following incubation, Alexa Fluor–conjugated secondary antibodies (1:500; Invitrogen Molecular Probes) were utilized to detect human antigens, and the sections were counterstained with DAPI (BD Pharmingen™-#564907). Subsequently, the processed slides were imaged by a confocal microscopy.

## Luciferase reporter assay

Cells were first seeded onto a 12-well plate at a density of $1.5 \times 10^5$ per well and grown overnight. The next day, the cells were co-transfected with 200 ng psicheck2 dual luciferase report vector containing WT/mutated CTSB 3'UTR sequence, 40 nM miR−218-1-3p mimics/negative

control (NC) mimics and 200 ng pLK reference plasmid using Lipofectamine™ 3000 transfection agent (ThermoFisher Scientific-#L3000015). The luciferase activity in each well was determined by using Dual-Glo® Luciferase Assay System (Promega-#E2920) 24 h post-transfection. For the EZH2 inhibitor treatments, cancer cells were co-transfected with 200 ng psicheck2 dual luciferase report vector containing WT/mutated CTSB 3'UTR sequence and 200 ng pLK reference plasmid using Lipofectamine™ 3000 transfection agent (Thermo-Fisher Scientific-#L3000015) in the presence or absence of 15 nM PF-06726304.

### In vivo tumorigenicity assay

Male nude mice aged 4–6 weeks were obtained from Slac Laboratory Animal Co. Ltd. (Hunan, China). After their acquisition, the mice were allowed a one-week acclimatization period at the animal facility of Sun Yat-sen University. During the experiments, they were maintained under specific conditions, including an ambient temperature of 22–24 °C, humidity control between 40% and 70%, and a 12-h light/dark cycle. They were provided with access to sufficient food and water. One week later, $2 \times 10^6$ WB$^{Ctrl}$ or WB$^{BMII}$ cells were first resuspended in 100 μL PBS-Matrigel mixture (PBS: Matrigel at 1:1) follow by injected into the lateral thigh of nude mice subcutaneously. 10 days later, the tumor bearing mice were then treated with 50 mg/kg CTSB inhibitor CA-074 or placebo via intraperitoneal injection daily for up to 20 days. The tumor sizes were measured by using a calliper every five days for up to 20 days. At the experimental end point, the mice were sacrificed and the tumors were dissected, photographed, measured, weighted and fixed for further analysis. The tumor volume (V) was calculated as $V = (Length \times width^2) \times 0.52$. The ethical committee permits a maximum tumor burden of 3000 mm³ or a 10% body weight loss, and none of the animal experiments conducted in this study exceeded this limit. For the tumorigenicity assay, 5000 or 10,000 WB$^{Ctrl}$/WB$^{BMII}$ cells were resuspended in 100 μL PBS-Matrigel mixture (PBS: Matrigel at 1:1), and injected subcutaneously into nude mice. 10 days post-injection, the tumor sizes were measured every five days for up to 10 weeks.

### Surgical orthotopic liver/spleen cancer model

$2 \times 10^6$ WB$^{Ctrl/BMII}$ cells (with or without stable GFP overexpression) or stable miR-218-1-3p mimic/NC mimic expressing WB$^{BMII}$ cells were first harvested and resuspended with 30 μL PBS mixed with Matrigel matrix (BD Biosciences) at 1:1 ratio, which were subsequently injected into either the Glisson 's capsules of livers or spleens of 4–6 weeks old male nude mice using an insulin syringe (BD Biosciences). For the CTSB inhibitor experiment, the orthotopic tumor bearing mice (i.e., 10 days post orthotopic liver implantation) were treated with placebo or 50 mg/kg CTSB inhibitor CA-074 via intraperitoneal injection daily for up to 18 days. The body weight and behavior of all mice was monitored every three days. 4-week post-injection, the mice were anesthetized and subjected to MRI according to the manufacturer's instruction. Briefly, the MRI analysis was performed on an M3 compact MRI system (Aspect Imaging, Israel) using Body Surface Coils. Mice were anaesthetized with a 2% isoflurane–oxygen mixture followed by placed prone in a holder and kept anesthetized with 1.5% isoflurane–oxygen via facial mask. The body temperature of nude mice was maintained using circulating water. All anatomical reference images were acquired and stored by using a T1-weighted Spin Echo (SE) sequence (TE/TR = 10/330, FOV = 30 × 30 mm, Matrix = 256 × 256, NEX = 10, Res.195 um, Acq Time 7:09 min:sec), and the slice thickness was 2.0 mm, with 0.1 mm gap. The MRI results were then interpreted by physicians. The blood samples were extracted from the mice and used for blood liver function tests. At the experimental end point, mice were sacrificed, and their livers/spleens/tumors excised, while the tumor weight was measured and recorded. All the harvested tissues were subsequently fixed in formalin at room temperature for overnight and paraffin

embedded, and then subjected to routine histopathological examination. The bile duct invasion and BDTT was assessed by performing serial sectioning of paraffin-embedded liver tissues whereby the entire liver was sectioned. The serial paraffin-embedded sections of the liver tissues/tumors (i.e., every 5 sections, ~25 μm separation) were stained with hematoxylin and subjected for IHC analysis.

### Proteomics analysis

In this study, protein samples were processed and analyzed by using a Orbitrap Exploris 480 mass spectrometer (ThermoFisher Scientific) with data dependent acquisition mode. The Proteome Discoverer software suite (version 2.3, ThermoFisher Scientific) was employed to perform the peptide identification and quantitation of our samples, as described previously[45]. Here's the detailed procedure: for sample preparation, WB$^{Ctrl}$ and WB$^{BMII}$ cells were cultured in a 100-mm culture plate at 37 °C in a 5% $CO_2$. Once the cells reached confluence, they were harvested by scraping with 800 μL of RIPA buffer. The cell lysates were then subjected to centrifugation at 4 °C for 30 min at 15,000×g, and the resulting supernatant was collected. The protein content of the samples was determined using the Pierce™ BCA Protein Quantification Kit (ThermoFisher Scientific-#23250). The cellular protein extracts were then subjected to sequential in-solution digestion with trypsin. In this process, the samples were initially precipitated with acetone and subsequently resuspended in a 50 mM urea buffer. Following this, they were reduced with dithiothreitol (DTT, 2 mM) at 37 °C for 1.5 h, alkylated with iodoacetamide (IAM, 10 mM) at 25 °C for 40 min, and finally diluted to 60 mM urea for an overnight digestion with trypsin (Sequencing Grade Modified Trypsin, Promega-#V5111) at 37 °C. The samples were then diluted 2-fold and subjected to another overnight digestion with trypsin at 37 °C. Subsequently, the resulting tryptic peptides were desalted using an HLB column (Oasis HLB, Waters-#186000383) and evaporated to dryness. A total of 300 μg of the dried protein digest was reconstituted in 10 μL of 0.1% formic acid in $H_2O$. Mass spectrometry analysis was then performed in positive ionization mode using an easy nano flex (Nanospray ionization, NSI) with a spray voltage set at 2.3 kV and a source temperature of 320 °C. The instrument operated in data-dependent acquisition mode, conducting full MS scans over a mass range of m/z 300–2000. During each cycle of data-dependent acquisition analysis, the most intense ions above a threshold ion count of 2.2e4 were selected for fragmentation at a normalized collision energy of 30% (HCD). The number of selected precursor ions for fragmentation was determined by the "Top Speed" acquisition algorithm, with a dynamic exclusion of 60 s. Fragment ion spectra were acquired either in the linear ion trap (IT) or the Orbitrap (OT) with a resolution of 60,000 depending on the method. The ion trap (IT) MS2 detection had an AGC of 4.0e5 and a maximum injection time of 50 ms, while the Orbitrap (OT) MS2 detection had an AGC of 5.0e4 and a maximum injection time of 50 ms. All data were acquired using Xcalibur software version 4.0. Peptides were injected into a nano-UPLC system (EASY-nLC 1200 liquid chromatograph) equipped with a 50-cm C18 column (EASY-Spray; 75–75-μm, PepMap RSLC C18, 2–6 μm particles, 45 °C). The chromatographic gradient had a length of 180 min, and the mobile phases used were as follows: Mobile Phase A ($H_2O$: HCOOH, 100:0.01) and Mobile Phase B (ACN: $H_2O$: HCOOH, 80:20:0.01). The gradient elution program included the following steps: 0–5 min, 4–8% B; 5–152 min, 8–28% B; 152–172 min, 28–38% B; 172–175 min, 38–100% B; 175–180 min, 100% B, followed by column equilibration with Mobile Phase A. For peptide identification and quantitation, the Proteome Discoverer software suite (version 2.3, ThermoFisher Scientific) was used. The data were searched against the Swiss-Prot protein sequence database with a precursor ion mass tolerance of 7 ppm at the MS1 level and allowance of up to three missed cleavages. The fragment ion mass tolerance was set to 20 ppm for the Orbitrap MS2 detection methods and to 0.5 Da for the linear ion trap MS2 detection methods. Variable modifications included oxidation of

methionine and N-terminal protein acetylation, while carbamido-methylation on cysteines was set as a fixed modification, as previously described[46].

## Trans-intrahepatic biliary epithelial migration assay

The upper compartment of a transwell insert (Corning) was first coated with Matrigel matrix (50 µg/mL, Corning) for 3 h, while HIBEpiC (i.e., a density of $2 \times 10^5$ cells/well) were evenly seeded onto the upper compartment of the transwell and left untouched for 24 h. After the cells adhered and formed a monolayer, the culturing medium was aspirated and the chamber was gently rinsed three times with PBS. A total of $2.5 \times 10^5$ human fluorescently-labeled HCC cells resuspended in 300 µL serum-free medium were seeded onto the upper compartment -/+ cathepsin B inhibitor CA-074 (APEXBIO-#A1926), and the bottom compartment was loaded with 700 µL complete medium containing 20% FBS for 48 h. For the cell counting process, four random microscopic fields were counted in each group using a fluorescent microscope, and the experiments were repeated three times independently.

## ELISA assay

The concentration of secreted/serum CTSB was determined by using human/rat CTSB ELISA kit (Abcam-#ab119584 (human); Elabscience-#E-EL-R2414c (rat)). For the preparation of culturing medium sample, $1 \times 10^6$ cells/well were plated into 6-well plates and cultured overnight. After washing three times with PBS, the cells were then cultured in 1.2 mL serum-free medium for 36 h. Finally, the culturing medium was carefully collected, centrifuged and stored in −80 °C freezer for later use. For the preparation of patient serum sample, HCC patients' venous blood samples were harvested and collected in serum separator tubes. After the clot formation procedure, the blood samples were then centrifuged at $2000 \times g$ for 10 min, while the serum samples were carefully collected and stored in a −80 °C freezer until use. For the BMI1 inhibitor experiments, the cells were treated with either DMSO, 1/10 µM PCT209 or 25/50 µM PRT4165 for 24 h before subjected to CTSB ELISA analysis.

## CTSB activity measurement assay

$WB^{Ctrl/BMI1}$, $PLC^{Ctrl/BMI1}$ or MHCC97H-shBMI1-1/−2/MHCC97H-scr cells were used to perform CTSB activity measurement following the manufacturer's instruction (Abcam-#ab65300). In brief, $6 \times 10^5$ cells/well were added into 6-well plates and grown for 48 h. Later, the cells were lysed with CB Cell Lysis Buffer, and the supernatants were collected and centrifuged at 12,000 g for 20 min at 4 °C. Afterwards, the protein concentration of each sample was measured by using a BCA Protein Assay Kit. 50 µL of the cell lysate containing the same amount of protein and 50 µL CB reaction buffer was added to each well, respectively. Finally, 2 µL CB substrate was added to each reaction and incubated at 37 °C for 1.5 h. The fluorescence signal (Ex/Em: 400 / 505 nm) was read and measured by using a Tecan Spark® 10 M Multimode plate reader. For the BMI1 inhibitor experiments, the cells were treated with either DMSO, 1/10 µM PCT209 or 25/50 µM PRT4165 for 24 h before subjected to CTSB activity measurement.

## Prediction of the potential CTSB targeting miRNAs

The miRNAs potentially regulated CTSB expression was predicted by using three different online search tools, including TargetScan[39] (https://www.targetscan.org/vert_80/), miRmap[47] (https://mirmap.ezlab.org/) and RNA22[48] (https://cm.jefferson.edu/rna22/), according to the web tool developers' instructions.

## Chromatin immunoprecipitation experiment

ChIP assay was carried out according to the instruction provided with the ChIP assay kit (SimpleChIP® Enzymatic Chromatin IP Kit, Cat# 9003, Cell Signaling Technology) according to previous published method with some modifications[44]. In brief, the cells were first crosslinked with 1% formaldehyde for 10 min at RT and then quenched with 0.125 mol/L glycine. Afterwards, chromatin was sonicated in the lysis buffer to 100−300 bp DNA fragments. Immunoprecipitation was performed using an anti-BMI1 antibody (Cell Signaling Technology-#6964), anti-H3K27me3 antibody (Cell Signaling Technology-#9733) or a normal Rabbit IgG (Cell Signaling Technology-#2729). The immunoprecipitated DNA was finally harvested from the beads and analyzed by PCR. The promoter primer sequences of the PRE sites for miR-218-1-3p promoter were given as follows: PRE site 1 forward: 5′-CAAAGGAAGGGTTAAACGGAAGG-3′, reverse: 5′-CGCTTACAGACA-CACACGTG-3′; PRE site 2 forward: 5′- CTAGAAGGAGAAGGACTACC-3′, reverse: 5′-TTCCCAGGCTTTGAATTCCG- 3′. For the EZH2 inhibitor experiments, $PLC^{BMI1}$ cells were first treated with or without 15 nM PF-067263004 for 24 h before conducting ChIP assays as described above.

## Fluorescence in situ hybridization (FISH) experiment

Fluorescence in situ hybridization was carried out as previously described[17]. Briefly, FISH of the miR-218-1-3p was conducted on paraffin fixed human HCC tumor tissue sections as indicated in the figure legends, while it was performed by using miRNA fish kit A (Gene-Pharma, China) and Cy3-labeled miR-218-1-3p fluorescence probe (GenePharma, China), and Cy3-labeled negative control (NC-Cy3) and positive control (U6-Cy3) probes were also used in the experiment. The intensity level of the staining in each tissue section was measured in 4 random microscopic fields using Image Pro Plus version 6.0 software.

## Statistics and Reproducibility

Unpaired two-tailed $t$ test, two-sided Chi-square test, two-sided Pearson correlation coefficient, log-rank (Mantel-Cox) test, one-/two-way ANOVA followed by Tukey's post hoc testing were used for statistical analysis in this study according to the data types. The information about statistical details (i.e., error bars represented standard error of the mean (SEM)) and methods, and also $n$ number was stated in the figure legends. The calculations of all statistical values were done using Graphpad Prism Version 8.0.1. Differences between groups with a $p$ value < 0.05 were considered statistically significant. In this study, the sample size was not predetermined using a statistical method and no data were excluded from the analyses. In the animal experiments, mice of similar ages and weights were randomly allocated to different experimental groups, and each group received the respective treatments as specified. The researchers conducting the animal experiments were not blinded since they needed to administer the specific treatments to the mice. Nevertheless, they maintained blinding during the subsequent data analysis. In experiments other than those involving animals, the samples/cells were randomly distributed into various experimental groups before treatment or observation. During microscopy and data collection using objective instruments, the researchers were not blinded to the group allocations, as it was essential to know which group each set of raw data corresponded to. However, they were blinded during the subsequent data analysis.

## Reporting summary

Further information on research design is available in the Nature Portfolio Reporting Summary linked to this article.

# Data availability

The publicly available mRNA sequencing and clinicopathological data of patients with HCC used in this study are accessible in the Cancer Genome Atlas Program (TCGA) database under accession code phs000178.v11.p8[30]. The use of these publicly available data from HCC was also consulted on the website (http://kmplot.com/analysis/index.php?p=service&cancer=liver_rnaseq) according to the web tool's instruction[31], under the specific product name: KM Plotter-Liver

Cancer. The proteomics data generated in the study has been deposited tin the Proteome X consortium via the PRIDE partner repository under accession code PXD037074 (https://www.iprox.cn//page/project.html?id=IPX0005126000). MiRNAs potentially regulating CTSB expression were predicted using three different online search tools: TargetScan[39] (https://www.targetscan.org/vert_80/), miRmap[47] (https://mirmap.ezlab.org/), and RNA22[48] (https://cm.jefferson.edu/rna22/), following the instructions provided by the developers of these web tools.The remaining data are available within the Article, Supplementary Information or Source Data file. Source data are provided with this manuscript. Source data are provided with this paper.

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

## Acknowledgements

We would like to thank for Professor Kairbaan Hodivala-Dilke for her technical support and advice. This work was supported by the National Natural Science Foundation of China (82173229 to L.-B. X., 81920108028 to P.-P.W., 82272830 to P.-P.W., 81972255 to C.L., 82173195 to C.L.); the Guangdong Basic and Applied Basic Research Foundation (2021A1515010095 to L.-B.X.); Science and Technology Program of Guangzhou (2023A03J0700 to L.-B.X.); Sun Yat-sen University Clinical Research 5010 Program (2018008 to C.L.); Guangzhou Key Laboratory of Precise Diagnosis and Treatment of Biliary Tract Cancer (202201020375 to C.L.); Guangdong Science and Technology Department (2023B1212060013 to P.-P.W., 2020B1212030004 to P.-P.W.).

## Author contributions

L.-B. X., Y.-F. Q. and L.S. conduced majority of the experiments and analysed the data; L.S. and C.H. performed proteomics analysis; Q.X., X.-D.S., R.S., J.C., Z.S., X.J., L.S., X.G. and X.K. assisted with the animal experiments and clinical study; R.Z. and C.L. provided clinical samples; C.L. and P.-P.W. conceived the study, supervised the research, and wrote the manuscript.

## Competing interests

The authors declare no competing interests.
