## [Peer Review File · Nature Communications]

Reviewers' Comments:

Reviewer #1:

Remarks to the Author:

In the present manuscript, Xu et al. investigated the possible origin of bile duct tumor thrombosis (BDTT) in human hepatocellular carcinoma (HCC). The authors found that BMI1 expressing tumor initiating cells (BMI1-high TICs) can induce BDTT. In particular, BMI1 overexpression was sufficient to transform liver progenitor cells into BMI1-high TICs, which are characterized by increased trans-intrahepatic biliary epithelial migration ability by secreting lysosomal CTSB. Indeed, orthotopic liver implantation of BMI1-high TICs into mice generated tumors and induced CTSB-driven tumor thrombus formation. Mechanistically, BMI1 epigenetically up-regulated CTSB secretion in TICs by repressing miR-218-1-3p expression.

The study is novel, interesting, and provides intriguing and highly relevant insights into the molecular pathogenesis of human HCC. The experiments in general were properly planned and conducted, and the data support the conclusions drawn. Some issues should be addressed to increase the value of the study further. Specifically:

1. Immunohistochemistry pictures for the various proteins should be done by showing in the same figure the bordering area between tumor and the non-tumor compartments.
2. Experiments should be repeated, at least in vitro, using BMI1 soluble inhibitors.

Reviewer #2:

Remarks to the Author:

Bile duct tumor thrombosis (BDTT) is a common finding in HCC and is often associated with poor clinical outcome. Authors demonstrated that BMI1 is responsible for the development of BDTT through regulating miR-213-1-30/CTSB signaling. BDTT is an important yet understudied clinical finding especially its underlying mechanisms. This study has demonstrated an interesting BDTT mouse model but overall mechanisms need to be substantially strengthened.

1. Authors demonstrated that BMI1 expression is elevated in patients with BDTT. The rationale for starting the story with BMI1 is unclear.
2. Authors showed that BMI1 high tumor initiating cells expressed higher levels of liver cancer stemness markers. However, it is unclear why liver cancer stemness has to be associated with BDTT. The link is not strong.
3. Authors later demonstrated that orthotopic injection of hepatic progenitor cells (WB-rat) with BMI1 high expression showed BDTT in nude mice. WB cells were also injected into mice through intra-splenic injection. 17% of the mice with WB-BMI1 cells displayed enlarged gall bladders and bile ducts with BDTT. These models are interesting, but the origin of cells is rat, limiting the broader application and human relevance. Demonstration with human cell lines is suggested.
4. Using proteomics, authors showed that CTSB was upregulated in WB-BMI1 cells while CTSB inhibitor blocked BDTT and improved outcome of mice with tumors. Again, it is unclear at this stage how CTSB is directly associated with BDTT.
5. Finally, authors showed that BMI1 regulates CTSB through suppressing miR-218-1-3p which would otherwise interact with CTSB to suppress CTSB. Therefore, BMI1 would lead to increase of CTSB which promotes the invasive ability of cancer cells through bile duct epithelial layer. While the mechanisms seem sound, effects of miRNAs are very modest generally.
6. Also, how BMI1 suppresses miR-218-1-3p is not clearly studied. For example, how it regulates the histone modifications which led to miR-218-1-3p suppression needs to be demonstrated.
7. Rescue experiments have to be performed to confirm the signaling axis.

Reviewer #3:

Remarks to the Author:

Xu, Qin and co-authors describe in the manuscript "BMI1 high tumour initiating cells trigger hepatocarcinogenesis and undergo lysosomal cathepsin B driven bile duct invasion to form tumour thrombus" BMI1 tumour initiating cells as triggers for the bile duct tumour thrombosis (BDTT) displayed in some of the patients with advanced hepatocellular carcinoma (HCC). BDTT is a severe complication of some patients with advanced HCC. HCC patients with BDTT are often misdiagnosed

and mistreated because of its unknown cellular origin and pathogenesis thus, it is essential to increase our understanding about the biological drivers of this disease. This manuscript sheds light over some of the biological mechanisms controlling the transformation of liver progenitor cells into tumour initiating cells that facilitate the generation of primary tumours and the formation of BDTT. This is an elegant, well-thought through and well-executed piece of work. Authors provide several different experimental evidence to support their conclusions using a wide range of techniques and experimental animal models. This reviewer has some comments that I hope authors will find useful to improve some aspects of the manuscript.

Major comments:

- Data shown in Fig. 1E should be replicated in the cohort of patients with BDTT used in Fig. 1D to demonstrate whether this observation is relevant to BDTT patients. Comparison of percentage overall survival of BDTT patients with low BMI1 versus high BMI1 should be shown.
- Fig 1G: A dual staining showing colocalization of BMI1 with a TIC marker would be helpful to further support the authors findings.
- Data shown in Fig. 1L should be replicated in cohort of patients with BDTT used Fig. 1D to demonstrate whether this observation is relevant to BDTT patients. Comparison of percentage overall survival of BDTT patients with low TIC-low BMI1 versus high TIC-high BMI1 should be shown.
- Fig 1I: It is not clear whether the figure refers to gene or protein signatures as in Fig. 1H.
- Fig. 2: WB Ctrl and PLC Ctrl columns in blue should display standard deviation (SD) bar.
- Fig. 2C-D: SD should be shown in each timepoint unless authors are displaying one representative experiment. If that is the case, this should be stated in the figure legend.
- Fig S2: Scr and WB Ctrl columns should display SD bar.
- Fig. 3F. and Fig. 3J BMI1 staining in liver tissue of WBBMI1 orthotopic tumour bearing mice or WBBMI1 implanted mouse with jaundice needs to be performed to demonstrate tumour thrombi were arisen from the invasion of WBBMI1 cells from primary tumour sites into the lumen of bile ducts.
- Data shown in Fig.4Q should be replicated in cohort of patients with BDTT. Comparison of percentage overall survival of BDTT patients with low BMI1-low CtsB versus high BMI1 -high CtsB should be shown.
- Fig. 5: Ctrl and scramble columns should display SD bar.

Minor comments:

- In line 135 authors describe findings on TIC-related gene signature however, Fig. 1G and H refer to protein expression detected by IHC and only Fig. S1C refers to gene signature. This should be amended on the text.
- Fig 1D. legend is incorrect according to the text.
- Fig 4B: extracellular region is not displayed in the figure in red as a cellular process related to CtsB in the figure. BP, CC, MF meaning of should be stated in figure legend.
- Fig 6G: It is not clear whether the ELISA was performed in media or cell lysate.
- Fig 6G and I: X axis are not scaled properly and looked distorted.
- Manuscript should be revised for some typos such as "corelated" in line 350.

Reviewer #4:

Remarks to the Author:

In this manuscript, Xu et al explore how bile duct tumor thrombus (BDTT) develops in HCC. Using HCC patient samples, they find that the incidence of bile duct tumor thrombus is higher in cancers with greater BMI1 expression. Overexpression of BMI1 in different cell lines induced proliferation, migration, and tumor formation in subcutaneous injections. Injection of BMI1 overexpressing cells into Glisson's capsule lead to tumor formation and the presence of jaundice with BDTT at a rate of 40%. Using proteomics, the authors found that the protein level of CTSB was increased with BMI1 overexpression in vitro. Using CTSB targeted siRNAs or small molecule inhibitors, the authors could reduce in vitro invasion and prevent the development of jaundice and BDTT. Furthermore, informatic and ChIP identified miR-218-1-3p as repressed by BMI1 and this miR appeared to bind to the 3' UTR of CTSB. The authors found that injection of miR-218-1-3p into their BDTT model could reduce the BDTT. While the formation of BDTT is somewhat low in their model; which could

reduce its generalized use, the mechanism of BMI1/miR-218-1-3p/CTSB/ they describe is interesting and relevant to those who study HCC. I do have a few comments that I believe would help strengthen the manuscript and improve its clarity.

1. There are multiple grammatical errors and portions of writing that are of poor English. These should be corrected.

2. It was not clear to me how it was determined what constituted a BMI1High and BMI1Low tumor sample. In Figure 1A, the low sample seems to have no expression by IHC. Was this distinction performed based on IHC or RNA level? What cutoff was used?

3. Bile duct tumor thrombus is a relatively rare complication of HCC; yet, in the database from the authors there appears to be a high proportion of these patients (43 out of 194, Fig 1D). Is there a reason for this? It would be helpful to have the clinical characteristics (labs, demographics, etc) of these patients divided by the presence and absence of BDTT and explained how they were determined to have BDTT. Additionally, what is the total size of this cohort as in some survival curve figures there are less patients and some there are more patients. Is there a reason that some patients are being excluded?

4. Figure 4Q: Do patients have tumors that are BMI1high/CTSBlow and BMI1low/CTSBhigh and what happens to them?

5. It would be informative if the authors could label their WBBMI1 cells with a fluorescent reporter to see if the breaks in the bile duct wall are due to tumor invasion vs immune cells as their H and E stains seem to indicate a marked inflammatory presence. Alternatively, if the authors have a specific antibody for their cells, immunostaining could be used.

REVIEWER COMMENTS

Reviewer #1 - HCC, mouse models (Remarks to the Author):

In the present manuscript, Xu et al. investigated the possible origin of bile duct tumor thrombosis (BDTT) in human hepatocellular carcinoma (HCC). The authors found that BMI1 expressing tumor initiating cells (BMI1-high TICs) can induce BDTT. In particular, BMI1 overexpression was sufficient to transform liver progenitor cells into BMI1-high TICs, which are characterized by increased trans-intrahepatic biliary epithelial migration ability by secreting lysosomal CTSB. Indeed, orthotopic liver implantation of BMI1-high TICs into mice generated tumors and induced CTSB-driven tumor thrombus formation. Mechanistically, BMI1 epigenetically up-regulated CTSB secretion in TICs by repressing miR-218-1-3p expression.

The study is novel, interesting, and provides intriguing and highly relevant insights into the molecular pathogenesis of human HCC. The experiments in general were properly planned and conducted, and the data support the conclusions drawn. Some issues should be addressed to increase the value of the study further. Specifically:

We sincerely appreciate the reviewer's positive recognition of the novelty and significance of our research in the context of human hepatocellular carcinoma (HCC). It is gratifying to learn that our data have been found to support the conclusions we have drawn. We are committed to addressing the specific points raised by the reviewer in order to significantly enhance the value of our study. Your constructive feedback is invaluable to us, and we are eager to make the necessary improvements based on your suggestions.

1. Immunohistochemistry pictures for the various proteins should be done by showing in the same figure the bordering area between tumor and the non-tumor compartments.

We extend our gratitude to the reviewer for the valuable feedback provided. In accordance with the reviewer's suggestion, we have taken action to update the IHC images in figure 1G. The revised figure now portrays the expression of a range of proteins – such as BMI1, CD44, CD133, and SOX9 – within the outlined region (dotted line) at the interface between tumor and the non-tumor compartments. Please refer to the new figure 1G as well as the accompanying illustration below for a clearer depiction of these modifications.

Figure 1G. Representative IHC images of BMI1, CD44, CD133 or SOX9 stained tumor sections derived from HCC patient with low or high BMI1 expression. Dot lines indicate the border between tumor (T) and the non-tumor compartments (NTC).

2. Experiments should be repeated, at least in vitro, using BMI1 soluble inhibitors.

We extend our sincere appreciation for the perceptive feedback provided by the reviewer. To amplify the clinical relevance of our study, we have incorporated fresh in vitro evidence. This evidence substantiates that the utilization of soluble BMI1 inhibitors, such as PRT209 and PRT4165, effectively curtails processes including colony formation, migration, tumor sphere generation, and invasion potential in endogenous BMI1-expressing human HCC cells MHCC97H (refer to new supplementary figure 2C, E, G, I and K). Furthermore, our findings elucidate that pre-treatment with BMI1 soluble inhibitors mitigates the exacerbated influence of conditioned medium (CM) derived from MHCC97H on cancer cell invasion, in comparison to cancer cells exposed to CM obtained from MHCC97H pretreated with placebo (refer to new supplementary figure 2M). Significantly, our data strongly underscore that the employment of BMI1 inhibitors not only diminishes the secretion and activity of CTSB in BMI1-expressing MHCC97H cells (please refer to new supplementary figure 4C, D, F, H), but also collectively reveal that the utilization of BMI1 inhibitors effectively regulates HCC cell properties and their paracrine impact on cancer cell invasions and probably as a potential treatment method for HCC patient with BDTT. Additionally, these new results are consistent with our previous observation in MHCC97H cells transfected with BMI1 targeting shRNA-1/-2 (refer to new supplementary figure 2 and 4). In summary, these findings provide robust evidence that BMI1 plays a pivotal role in controlling cancer cell properties and stemness, while also contributing to the malignant transformation of liver progenitor cells into tumor-initiating cells. Please also see new supplementary 2 and supplementary figure 4 and figures below alongside with result section page 7, 8, 11, 12.

Related new supplementary figure 2 legend: (C) CCK8 proliferation assays of the MHCC97H cells after treated with DMSO or BMI1 inhibitors (PCT209 or PRT4165) at different doses over time. (E) Colony formation assay of MHCC97H cells after treated with DMSO, PCT209 or PRT4165 at indicated doses. (G) Transwell migration assay of MHCC97H cells after treated with DMSO, PCT209 or PRT4165 at indicated doses. (I) Transwell invasion assay of MHCC97H cells after treated with DMSO, PCT209 or PRT4165 at indicated doses. (K) Tumor sphere formation of MHCC97H cells after treated with DMSO, PCT209 or PRT4165 at indicated doses. (M) Transwell invasion assay of MHCC97H cells after exposed with CM from DMSO, PCT209 or PRT4165 pre-treated MHCC97H cells (n= 3 experimental repeats).

Related new supplementary figure 4 legend: (C, D) Western blot and RT-PCR analysis of the CTSB expression in MHCC97H cells treated with either placebo or BMI1 inhibitors (i.e. PCT-209 or PRT4165) at indicated doses. (F) ELISA assay analysis of conditioned medium harvested from MHCC97H after treated with either placebo or BMI1 inhibitors (i.e. PCT-209 or PRT4165) at indicated doses. (H) Measurement of the CTSB activity in conditioned medium derived from MHCC97H cells after treated with either placebo or BMI1 inhibitors (i.e. PCT-209 or PRT4165) at indicated doses (n= 3 experimental repeats).

Reviewer #2 - HCC Stem cells, TME (Remarks to the Author):

Bile duct tumor thrombosis (BDTT) is a common finding in HCC and is often associated with poor clinical outcome. Authors demonstrated that BMI1 is responsible for the development of BDTT through regulating miR-213-1-30/CTSB signaling. BDTT is an important yet understudied clinical finding especially its underlying mechanisms. This study has demonstrated an interesting BDTT mouse model but overall mechanisms need to be substantially strengthened.

We express our gratitude to the reviewer for displaying keen interest in our novel BDTT mouse model. In response to your inquiries, we have substantially fortified our work by furnishing additional comprehensive evidence. This augmentation has been particularly geared towards enhancing the elucidation of the comprehensive mechanisms underpinning BMI1-mediated BDTT.

1. Authors demonstrated that BMI1 expression is elevated in patients with BDTT. The rationale for starting the story with BMI1 is unclear.

We greatly appreciate the reviewer's insightful comment. As highlighted in the introduction section of our manuscript, it is important to note that both our research and that of other investigators have previously demonstrated BMI1's established role as a regulator of tumor-initiating cells, particularly in the context of cancer stemness, across diverse cancer types (Wang et al., Stem Cells Int 2020; Chen et al., Cell Stem Cell 2017; Wang et al., J Hematol Oncol 2016; Zhang et al., Oncogenesis 2016; Ma et al., Cancer Biomark 2018; Sparmann et al., Nature Reviews Cancer 2006; Park et al., Journal Clinical Investigation 2004). Furthermore, the significance of BMI1 extends to hepatocellular carcinoma (HCC) initiation and progression, as evidenced by studies in the field (Chiba et al., Hepatology 2006; Chiba et al., Cancer Research 2008). Furthermore, it is essential to highlight that our laboratory had previously conducted a preliminary study correlating BMI1 expression levels with the development of bile duct tumor thrombus (BDTT) in a small cohort of HCC patients. It's worth noting that, during that phase, our exploration of the mechanistic role was hindered by the scarcity of a substantial patient cohort with BDTT and a suitable animal model for studying BDTT (Zhang et al., Hepatogastroenterology 2013). Taking this context into account, the present study was initiated with a clear objective to delve deeper into the mechanistic implications of BMI1 in driving BDTT. To accomplish this, we have engaged a larger cohort of HCC patients, encompassing individuals both with and without BDTT. Additionally, we have leveraged our newly developed BDTT mouse model by performing orthotopic implantation of BMI1 overexpressing TICs into the livers of nude mice in order to augment our investigative approach. For your interest, we have taken the opportunity to provide a more detailed rationale for launching this investigation into BDTT's connection with BMI1 in the introduction section of the manuscript. For a more comprehensive understanding, we kindly direct your attention to the revised introduction on page 3. We are confident that this clarification effectively addresses the concerns raised by the reviewer.

2. Authors showed that BMI^{high} tumor initiating cells expressed higher levels of liver cancer stemness markers. However, it is unclear why liver cancer stemness has to be associated with BDTT. The link is not strong.

We extend our appreciation for the valuable input provided by the reviewer. As acknowledged, bile duct tumor thrombus (BDTT) constitutes an essential yet understudied clinical phenomenon, particularly concerning its intricate underlying mechanisms. In our pursuit to shed light on this enigma, we have dedicated significant efforts to unravel its intricacies. Our primary objective is to identify potential therapeutic interventions for hepatocellular carcinoma (HCC) patients confronted with the challenge of BDTT. Existing literature indicates that liver cancer stemness markers, such as CD133, CD44, and SOX9, are closely associated with HCC initiation and progression (Zhu et al., International Journal of Cancer 2010; Liu et al., Hepatology 2016). Cells expressing elevated levels of these markers are known to possess heightened migration and invasion capabilities (Prager et al., Cell Stem Cell 2019). Coincidentally, preliminary studies have suggested that tumor-initiating cells (TICs) originating from malignantly transformed liver progenitor cells might infiltrate the sub-epithelial layer of adjacent bile duct walls, thus forming tumor thrombus (Zhang et al., Medicine (Baltimore) 2015). Concurrently, BMII's role in governing TIC self-renewal has been highlighted (Wang et al., Stem Cells Int 2020; Bansal et al., Clinical Cancer Research 2016). Our research substantiates these observations by showcasing that BMII overexpression induces the transformation of rat liver progenitor cells (referred to as WB) into TICs. These TICs exhibit an elevated expression of liver cancer stemness markers and demonstrate the ability to initiate tumorigenesis in vivo (refer to figure 2P-Q). We further elucidate the connection by demonstrating that orthotopic injection of BMII-expressing TICs (WB^{BMII}) initiates primary tumor formation and BDTT in both liver and splenic models. In contrast, the introduction of empty vector-transfected WB cells (WB^{ctrl}), characterized by a low stem cell marker signature including CD44, CD133, and SOX9, fails to initiate tumor and BDTT formation (refer to figure 3 and new supplementary figure 3). Importantly, our clinical studies also showed that the expression of BMII and TIC-related signature was positively correlated and their expression associated with the incidence of BDTT as well as patient prognosis (refer to figure 1 and new supplementary figure 1).

Additionally, our newly established in vitro model reinforces these findings. Cells overexpressing BMII, encompassing human HCC cells and WB^{BMII} cells, exhibit enhanced trans-intrahepatic epithelial migration capabilities (refer to figure 5G-J). Conversely, BMII depletion in human HCC cells MHCC97H, characterized by high endogenous BMII levels (refer to figure 4C), demonstrates the opposite effect (refer to supplementary figure 5K). To further underscore our observations, we present new additional evidence indicating that BMII depletion in MHCC97H cells reduces the expression of CD44, CD133 and SOX9 (refer to new supplementary figure 2P). Collectively, these discoveries, in conjunction with our new additional evidence, indicate that BMII-mediated cancer stemness plays a pivotal role in both tumor initiation and the subsequent formation of BDTT. For further elaboration, please refer to the detailed results section page 8-10, 12-13 and the discussion section of the manuscript page 17-18.

Related new supplementary figure 2 legend:(P) RT-PCR analysis of the expression of CD44, CD133 and SOX9 in MHCC97H transfected with BMI1 targeting shRNA-1/-2 or scramble shRNA (n= 3 experimental repeats).

3. Authors later demonstrated that orthotopic injection of hepatic progenitor cells (WB-rat) with BMI high expression showed BDTT in nude mice. WB cells were also injected into mice through intra-splenic injection. 17% of the mice with WB-BMI1 cells displayed enlarged gall bladders and bile ducts with BDTT. These models are interesting, but the origin of cells is rat, limiting the broader application and human relevance. Demonstration with human cell lines is suggested.

We extend our gratitude for the positive feedback from the reviewer regarding our BDTT model. Unfortunately, neither commercially available human liver progenitor cell lines nor relevant references in the literature are at our disposal to replicate the rat liver progenitor cell experiment. While we have been diligently attempting to cultivate endogenous BMI1-expressing human HCC cell line Huh7 in a three-dimensional culture to enhance their aggressiveness, consistent generation of orthotopic liver tumors upon injection into nude mice has proven challenging, which is consistent with previous publication (Wu et al., Sci Rep 2016; 6: 35230).

Despite these limitations, we firmly uphold that the utilization of rat liver progenitor cells has furnished substantial evidence to support our concept. Our research compellingly underscores the significance of BMI1-mediated malignant conversion of liver progenitor cells into tumor-initiating cells, significantly contributing to liver tumor development and BDTT formation in two different animal models (refer to figure 2 and 3). While we acknowledge the ideal scenario would involve human cell lines, we firmly believe that our current model substantially contributes to advancing our understanding of the mechanisms underlying BDTT. Please see figure 2, 3, 5 and 6 for more details as well as discussion section page 17.

4. Using proteomics, authors showed that CTSB was upregulated in WB-BMI1 cells while CTSB inhibitor blocked BDTT and improved outcome of mice with tumors. Again, it is unclear at this stage how CTSB is directly associated with BDTT.

We express our appreciation for the reviewer's insightful comment. It's worth noting that the exploration of the underlying mechanism of BDTT in the context of HCC is an uncharted area within the research field. Our study thus stands as one of the pioneering investigations in this direction. Through our research, we demonstrate that orthotopic liver implantation of BMI1^{High} tumor initiating cells (TICs) instigates tumorigenesis and BDTT. To unravel the potential mechanism behind BMI1-mediated BDTT, we conducted proteomics analysis on WB^{Ctrl} and

WB^{BMII} cells. The outcomes of this analysis reveal the up-regulation of lysosomal cathepsin B (CTSB), a member of the lysosomal cysteine protease family, in BMII overexpressing cells (WB^{BMII}). This observation is substantiated by GO enrichment analysis, which highlights significant enrichment in CTSB-related cellular pathways and components, including the extracellular region, cell adhesion, and cell junction, in WB^{BMII} cells (Please refer to figure 4A and B). Further experimental results reinforce these findings, showcasing the up-regulation of CTSB expression and secretion in BMII-expressing human HCC cells (Please refer to figure 4D-I, supplementary Figure 4). Notably, as suggested by the reviewer 1, we provide new data indicating that depletion of BMII using siRNA or inhibitors leads to reduced CTSB secretion and activity in human HCC MHCC97H cells (Please refer to new supplementary figure 4C, D, F and H). From a clinical perspective, we have unveiled correlations between CTSB expression and patient survival, BDTT incidence, BMII expression, and tumor-Initiating cell (TIC) gene signatures in HCC patients (refer to figure 4J-T). We have now also provided new supplementary clinical data derived from the TCGA database to support our findings, indicating that high CTSB expression also correlates with poor overall survival in HCC patients (Please refer to new supplementary figure 4I). Importantly, we provide new co-immunostaining experiments indicating that BMII overexpressing cells invaded into the bile duct to form BDTT (refer to new supplementary 3A). In additionally, we perform new cell tracing experiments by orthotopically injecting GFP expressing WB^{BMII} cells into the livers of nude mice, demonstrating that GFP tagged BMII^{high} TICs induce BDTT via direct invasion into bile ducts in our animal models (refer to new supplementary figure 3B), which is consistent with our existing observation. In vivo, treatment with CTSB inhibitors curtails BDTT formation and extends mouse survival in BMII^{high} TICs orthotopically implanted mice (Please refer to figure 5M-Q). Additionally, depletion of CTSB using inhibitors or siRNAs reduces the invasion and trans-intrahepatic migration capabilities in BMII-expressing cells (Please refer to figure 5G-I). Notably, the overexpression of CTSB in BMII-depleted HCC cells rescues their invasion and trans-intrahepatic migration abilities (Please refer to supplementary figure 5J and K). Collectively, our comprehensive experiments substantiate that CTSB plays a direct role in BDTT formation, and BMII-expressing TICs secrete CTSB to facilitate their invasion into bile ducts. This multifaceted investigation significantly contributes to advancing our understanding of the intricate mechanisms underlying BDTT. Please also see result section page 10-13, discussion section page 18-19, figure 4-5, supplementary figure 3-5 for more details.

Related new supplementary figure 4 legend:(I) Correlation study between CTSB expression and overall survival in HCC patients (n= 364 patients, TCGA database).

Related new supplementary figure 3 legend: (A) Representative immunofluorescent images of BMI1 and CK7 stained tumor sections derived from WB^{Ctrl} or WB^{BMI1} implanted mice are given. (B) Provided are representative immunofluorescent images capturing the GFP overexpressing BMI1 cells and CK7 antibody staining within tumor sections extracted from nude mice implanted with GFP labelled WB^{Ctrl} or WB^{BMI1} cells.

5. Finally, authors showed that BMI11 regulates CTSB through suppressing miR-218-1-3p which would otherwise interact with CTSB to suppress CTSB. Therefore, BMI1 would lead to increase of CTSB which promotes the invasive ability of cancer cells through bile duct epithelial layer. While the mechanisms seem sound, effects of miRNAs are very modest generally.

We would like to thank for the reviewer's comment. In our study, we have presented compelling evidence demonstrating that miR-218-1-3p overexpression results in a substantial reduction in CTSB expression and secretion within BMI1 overexpressing cells, including WB^{BMI1} and PLC^{BMI1} cells (refer to figure 6F and G, supplementary figure 6F and G). Additionally, miR-218-1-3p overexpression exerts a strong inhibitory impact on the invasion and trans-intrahepatic biliary epithelial migration capacity of WB^{BMI1} and BMI1-overexpressing PLC cells as well as MHCC97H cells in vitro (refer to supplementary figure 6K-M). Furthermore, our findings extend into in vivo experiments, where miR-218-1-3p's influence is evident. Its overexpression leads to a reduction in the incidence of jaundice and BDTT formation in WB^{BMI1} orthotopically implanted mice, all the while not affecting primary tumor growth (refer to figure 6K-N and supplementary figure 6N and 7). We acknowledge the cautious evaluation of miRNA effects; nevertheless, our comprehensive analyses provide robust support for the significant role of miR-218-1-3p in modulating CTSB and its consequential impact on cancer cell invasiveness and trans-intrahepatic biliary epithelial migration. Please also see result section page 13-16 and discussion section page 19, figure 6 and new supplementary figure 6 and 7.

6. Also, how BMI1 suppresses miR-218-1-3p is not clearly studied. For example, how it regulates the histone modifications which led to miR-218-1-3p suppression needs to be demonstrated.

We would like to thank for the reviewer's comment. Since our result indicates that BMI1, a known regulator of miRNA expression (Cao et al., Cancer Cell 2011; Chen et al., Aging 2021), modulates CTSB expression at post-translational level, we therefore use CTSB mRNA sequence to predict its regulatory miRNAs using three different online search tools. Further Venn diagram analysis indicates that 41 of these predicted miRNAs are commonly identified by all these tools, while miR-218-1-3p is the only miRNA expression of which is correlated with BMI1 expression (refer to figure 6A-C). Consistently, previous studies suggest that CTSB expression is potentially regulated by miRNAs (Venkataraman et al., J Biol Chem 2013). Mechanistically, BMI1 has been shown to silence gene/miRNA expression by collaborating with EZH2 to increase the trimethylation of histone 3 at lysine 27 (H3K27meth) (Pietersen et al., Breast Cancer Research 2008; Chou et al., Cancer Research 2012; Benard et al., Plos One 2014). To further substantiate this finding in our study, we conducted new Western blot analysis, revealing an up-regulation of trimethylated histone 3 (H3K27meth) expression in BMI1-overexpressing PLC cells (PLC^{BMI1}) (refer to new supplementary figure 6A). Notably, the level of H3K27 methylation decreased in these cells following treatment with the EZH2 inhibitor PF-06726304 (refer to new supplementary figure 6B). Additionally, our ChIP assays uncovered elevated occupancy of BMI1 and trimethylated H3K27 on the promoter's PRE sites of miR-218-1-3p in BMI1-expressing cells. However, treatment with EZH2 inhibitors diminished the augmented binding of H3K27me3 to these PRE sites seen in BMI1-expressing cells (refer to figure 6D and E, and new supplementary figure 6C). Significantly, our new data showed that EZH2 inhibitor treatment enhanced the expression of miR-218-1-3p in BMI1-overexpressing PLC cells and MHCC97H cells respectively (refer to new supplementary figure 6D). Furthermore, it decreased the luciferase activity in cells transfected with a luciferase vector containing the WT-CTSB 3'UTR sequence but had no impact on those with the mutated CTSB 3'UTR sequence (refer to new supplementary figure 6H). These new collective findings strongly suggest that BMI1 collaborates with EZH2 to suppress miRNA-218-1-3p expression by elevating histone trimethylation at lysine 27, thereby enhancing CTSB expression. For a comprehensive understanding, please also refer to figure 6A-C, new supplementary figure 6A-D and H, and the figure below in the results section on page 13-15, and discussion section page 19.

Related supplementary figure 6 legend:(A) Western blot analysis of the levels of H3K27me3 and total H3 in empty vector transfected PLC (PLC^{Ctrl}) and BMI overexpressing PLC (PLC^{BMI1}) cells. β -tubulin was used as loading control. Molecular weight markers are given on the right-hand side. (B) Western blot analysis of the levels of H3K27me3 and total H3 in PLC^{BMI1} cells treated with or without EZH2 inhibitor PF-06726034. (C) ChIP analysis of H3K27me3 binding to the PRE1 and PRE2 sites in the miR-218-1-3p promoter in PLC^{BMI1} cells, with or without EZH2 Inhibitor PF-06726034 treatment (n= 3 experimental repeats). (D) RT-PCR analysis of the level of miR-218-1-3p in PLC^{BMI1} and MHCC97H cells treated with or without PF-06726304.(H) Luciferase reporter assays of PLC^{BMI1} or MHCC97H cells after co-transfected with a luciferase reporter vector containing the sequence of either predicted wild type-3'-UTR (WT-CTSB) or mutated-3'-UTR region (Mut-CTSB) of CTSB and pLK reference plasmid in the presence or absence of EZH2 inhibitor PF-06726034 treatment (n= 3 experimental repeats).

7. Rescue experiments have to be performed to confirm the signaling axis.

We extend our gratitude for the reviewer's feedback. It is important to acknowledge that we have extensively conducted a series of both in vitro and in vivo rescue experiments using WB^{BMI1} cells and human HCC cell lines, systematically validating the proposed signaling axis throughout the manuscript (please refer to figure 5 and 6, as well as new supplementary figure 2, 4, 5-7). In response to the reviewer's valuable suggestion, we have incorporated new evidence that highlights the effectiveness of soluble BMI1 inhibitors in inhibiting colony formation, tumor sphere generation, migration, and invasion (please see new supplementary figures 2C, E, G, I, K). Additionally, we demonstrate that conditioned medium (CM) derived from BMI1 inhibitor-pre-treated MHCC97H cells no longer enhances cancer cell invasion, in contrast to cells treated with CM from placebo-pre-treated MHCC97H cells (please refer to new supplementary figure 2M). Furthermore, these BMI1 inhibitors lead to a reduction in CTSB expression and activity in human HCC MHCC97H cells (refer to new supplementary figure 4C, D, F, H). Crucially, our manuscript already provides robust in vivo evidence illustrating that the depletion of CTSB through an inhibitor or miR-213-1-3p significantly reduces the incidence of BDDT in WB^{BMI1} orthotopically implanted nude mice (please refer to figure 5 and 6, as well as supplementary

figure 5-7). Furthermore, our research showcases how aberrant expression of *CTSB* or *miR-218-1-3p* modulates *BMII*-mediated cancer cell invasion and trans-intrahepatic epithelial migration *in vitro* (please refer to figure 5 and 6, as well as supplementary figure 5 and 6). In light of these comprehensive findings, we confidently assert that we have executed a substantial array of rescue experiments, effectively corroborating the pivotal role of the *BMII-miR-218-1-3p-CTSB* signaling axis in *BDTT*. Please see new supplementary figures 2C, E, G, I, K M, new supplementary figure 4C, D, F, H, figure 5 and 6 as well as supplementary figure 5-7, the figures below and result section page 12-16, discussion section page 18-20.

Related supplementary figure 2 legend: (C) CCK8 proliferation assays of the *MHCC97H* cells after treated with DMSO or *BMII* inhibitors (*PCT209* or *PRT4165*) at different doses over time. (E) Colony formation assay of *MHCC97H* cells after treated with DMSO, *PCT209* or *PRT4165* at indicated doses. (G) Transwell migration assay of *MHCC97H* cells after treated with DMSO, *PCT209* or *PRT4165* at indicated doses. (I) Transwell invasion assay of *MHCC97H* cells after treated with DMSO, *PCT209* or *PRT4165* at indicated doses. (K) Tumor sphere formation of *MHCC97H* cells after treated with DMSO, *PCT209* or *PRT4165* at indicated doses. (M) Transwell invasion assay of *MHCC97H* cells after exposed with CM from DMSO, *PCT209* or *PRT4165* pre-treated *MHCC97H* cells.

Related supplementary figure 4 legend: (C, D) Western blot and RT-PCR analysis of the CTSB expression in MHCC97H cells treated with either placebo or BMI1 inhibitors (i.e. PCT-209 or PRT4165) at indicated doses. (F) ELISA assay analysis of conditioned medium harvested from MHCC97H after treated with either placebo or BMI1 inhibitors (i.e. PCT-209 or PRT4165) at indicated doses. (H) Measurement of the CTSB activity in conditioned medium derived from MHCC97H cells after treated with either placebo or BMI1 inhibitors (i.e. PCT-209 or PRT4165) at indicated doses (n= 3 experimental repeats).

Reviewer #3 - HCC, Cathepsin (Remarks to the Author):

Xu, Qin and co-authors describe in the manuscript “BMI1high tumour initiating cells trigger hepatocarcinogenesis and undergo lysosomal cathepsin B driven bile duct invasion to form tumour thrombus” BMI1 tumour initiating cells as triggers for the bile duct tumour thrombosis (BDTT) displayed in some of the patients with advanced hepatocellular carcinoma (HCC). BDTT is a severe complication of some patients with advanced HCC. HCC patients with BDTT are often misdiagnosed and mistreated because of its unknown cellular origin and pathogenesis thus, it is essential to increase our understanding about the biological drivers of this disease. This manuscript sheds light over some of the biological mechanisms controlling the transformation of liver progenitor cells into tumour initiating cells that facilitate the generation of primary tumours and the formation of BDTT. This is an elegant, well-thought through and well-executed piece of work. Authors provide several different experimental evidence to support their conclusions using a wide range of techniques and experimental animal models. This reviewer has some comments that I hope authors will find useful to improve some aspects of the manuscript.

We sincerely thank for the reviewer’s complement about our work. We appreciate that the reviewer found the conclusion of our work is strongly supported by several different experimental evidence obtained from a wide range of techniques and experimental animal works. We also believe that our work sheds light over some of the biological mechanisms controlling the transformation of liver progenitor cells into tumor initiating cells that facilitate the generation of primary tumors and the formation of BDTT.

Major comments:

- Data shown in Fig. 1E should be replicated in the cohort of patients with BDTT used in Fig. 1D to demonstrate whether this observation is relevant to BDTT patients. Comparison of percentage overall survival of BDTT patients with low BMI1 versus high BMI1 should be shown.

We want to express our sincere appreciation for the insights provided by the reviewer. The same cohort of patients was used for both figure 1D and 1E. The disparity in patient numbers arose from our initial decision to exclude individuals with low BMI1 expression and BDTT, as well as those with high BMI1 expression without BDTT. This cautious approach was taken to preempt any potential confusion and is the primary reason for the variation in patient counts between figure 1D and 1E. In response to your valuable feedback, we have thoughtfully incorporated the data concerning HCC patients without BDTT but with high BMI1 expression, and conversely, patients with BDTT and low BMI1 expression. It's important to note that the updated results contrast with HCC patients who exhibit high BMI1 expression and BDTT, or low BMI1 expression without BDTT. Our revised findings highlight that the association between low BMI1 expression in HCC patients and BDTT, or high BMI1 expression without BDTT, lacks significance and involves a smaller sample size in the context of overall patient survival.

For a comprehensive visual representation of these refined results, we invite you to refer to the revised figure 1E and the corresponding figure below, which is available in the results section on page 6.

Related figure 1 legend:(E) Among HCC patients, those with both BDTT and high BMI1 expression experienced notably poorer overall survival when compared to patients without BDTT but with low BMI1 expression. HCC patients exhibiting BDTT alongside low BMI1 expression did not show a significant disparity in overall survival in comparison to the patients without BDTT but possessing high BMI1 expression (n=194 HCC patients, our cohort).

• Fig 1G: A dual staining showing colocalization of BMI1 with a TIC marker would be helpful to further support the authors findings.

We extend our gratitude for the insightful suggestion provided by the reviewer. In response, we have conducted co-staining experiments of BMI1 with CD133, CD44, or SOX9 in tumor sections. The outcomes of these new experiments robustly validate the co-expression of BMI1 with CD133, CD44, and SOX9 in the HCC tumor sections. For a detailed visual representation of these findings, kindly refer to the newly added supplementary figure 1C and figure below and the result section page 6.

Related supplementary figure 1 legend (C) Representative immunofluorescent images of BMI1 and CD44/CD133/SOX9 stained tumor sections from HCC patient with either high or low BMI1 expression are given. Magnified pictures are provided on the right-hand side.

- Data shown in Fig. 1L should be replicated in cohort of patients with BDTT used Fig. 1D to demonstrate whether this observation is relevant to BDTT patients. Comparison of percentage overall survival of BDTT patients with low TIC-low BMI1 versus high TIC-high BMI1 should be shown.

We sincerely appreciate the insightful comment from the reviewer. We wish to emphasize that the same patient cohort was utilized for both figure 1D and 1L. The disparity in patient numbers arises from our initial decision to exclude individuals with a low TIC-related protein signature and high BMI1 expression, as well as those with a high TIC protein signature and low BMI1 expression. This exclusion was intended to avoid potential confusion in the analysis. In response to the reviewer's valuable feedback, we have now incorporated the data related to BDTT patients exhibiting a low TIC-related protein signature and high BMI1 expression, as well as those with a high TIC-related protein signature and low BMI1 expression, aligning with your suggestion. Importantly, the updated data strongly indicate that these particular patient subsets do not exhibit significant differences in overall survival when compared to each other. This contrasts with patients displaying the dynamics between low TIC-related protein signature and low BMI1 expression versus high TIC-related protein signature and high BMI1 expression. To gain a comprehensive understanding of these revised findings, we invite you to refer to the updated figure 1L and the corresponding content on manuscript page 7.

Related figure 1 legend: (L) Elevated expression of the TIC-related protein signature along with BMI1 was associated with a more unfavorable patient survival outcome when contrasted with patients exhibiting low TIC-related gene signature and BMI1 expression. However, no significant distinction in overall survival emerged between HCC patients with either low TIC-related gene signature and high BMI1 expression or those displaying high TIC-related gene signature and low BMI1 expression (n=194 HCC patients, our cohort).

• Fig 1I: It is not clear whether the figure refers to gene or protein signatures as in Fig. 1H. *We apologize for it. The figure refers to protein signatures in Fig. 1H and I. We have now made it clear in our figure and figure legend.*

• Fig. 2: WBCtrl and PLCtrl columns in blue should display standard deviation (SD) bar. *We thank for the reviewer's comment. In figure 2, we show the data of WB^{BMI1} and PLC^{BMI1} relative to the WB^{Ctrl} and PLC^{Ctrl} levels in three independent experiments. Therefore, the WB^{Ctrl} and PLC^{Ctrl} columns do not have error bars after the data normalization. It is done as described before in the study of Barczak et al., Nature Comms 2023 and Rho et al., Cell metabolism 2023. We have now made it clear in the figure legend.*

• Fig. 2C-D: SD should be shown in each timepoint unless authors are displaying one representative experiment. If that is the case, this should be stated in the figure legend. *We greatly appreciate the reviewer's input. We wish to clarify that we have indeed incorporated error bars for each timepoint in figure 2C and D. However, due to the minor variations between experiments, these error bars might not be distinctly visible on the graph. It's important to note that the experiment was independently repeated three times to ensure robustness and consistency of the results. Please also see the figure legend.*

• Fig S2: Scr and WBCtrl columns should display SD bar. *We thank for the reviewer's comment. In figure S2, we show the results of WB^{BMI1} after normalized to the scramble (scr) transfected cells or WB^{Ctrl} cells. Therefore, the Scr and WB^{Ctrl} columns do not have error bars after the normalization. It is done as described before in the study of Barczak et al., Nature Comms 2023 and Rho et al., Cell Metabolism 2023. We have now made it clear in the supplementary figure legend.*

• Fig. 3F. and Fig. 3J BMI1 staining in liver tissue of WBBMI1 orthotopic tumour bearing mice or WBBMI1 implanted mouse with jaundice needs to be performed to demonstrate tumour thrombi were arisen from the invasion of WBBMI1 cells from primary tumour sites into the lumen of bile ducts.

We would like to express our gratitude for the valuable input provided by the reviewer. In response to the recommendation, we have conducted additional co-immunostaining experiments, which clearly demonstrate that BMI1-overexpressing cells invade from primary tumor sites through the CK7-positive bile duct epithelial cell layer into the lumen of bile ducts, leading to the formation of BDTT (as shown in the new supplementary figure 3A). Furthermore, we have performed orthotopic injections of GFP-overexpressing WB^{ctrl} and WB^{BMI1} cells into the livers of nude mice respectively. Subsequently, we carefully collected and fixed the tumors from mice implanted with WB^{BMI1} cells, followed by co-immunostaining using GFP and CK7 antibodies. These latest findings conclusively confirm that GFP-positive WB^{BMI1} cells invade from the primary tumor sites into the CK-7 positive bile ducts (as illustrated in supplementary figure 3B). For a visual representation of these compelling results, we kindly invite you to refer to the newly included supplementary figure 3A and B, and the figure provided below, as well as the relevant details on page 9 and 10 in the results section.

Related supplementary 3 figure legend: (A) Representative immunofluorescent images of BMI1 and CK7 stained tumor sections derived from WB^{Ctrl} or WB^{BMI1} implanted mice are given. (B) Provided are representative immunofluorescent images capturing the GFP overexpressing BMI1 cells and CK7 antibody staining within tumor sections extracted from nude mice implanted with GFP labelled WB^{Ctrl} or WB^{BMI1} cells.

• Data shown in Fig.4Q should be replicated in cohort of patients with BDTT. Comparison of percentage overall survival of BDTT patients with low BMI1-low CtsB versus high BMI1 -high CtsB should be shown.

We appreciate the reviewer's comment. Unfortunately, within our HCC patient cohort with BDTT, there were very few patients exhibiting both low BMI1 and low CTSB expression. Given this limited sample size, conducting a meaningful comparison between BDTT patients with low BMI1 and low CTSB expression and those with high BMI1 and high CTSB expression would not yield statistically robust results. However, we have already demonstrated that in HCC patients, high BMI1 and CTSB expression is significantly associated with poorer overall survival when compared to patients with low BMI1 and CTSB expression, while their expression was positively correlated (please refer to the new figure 4Q and figure 4R for more details). Additionally, we already showed that high tumor expression level of CTSB was correlated with increased incidence of BDTT (refer to figure 4N and O), while the serum level of CTSB was up-regulated in HCC patients with BDTT (refer to figure 4P). More importantly, we have included new clinical data demonstrating a correlation between high CTSB expression and BDTT with poorer overall survival in HCC patients, as opposed to patients with low CTSB expression and BDTT (please refer to new supplementary figure 4J). These findings provide compelling evidence for the association between BMI1 and CTSB expression and the formation of BDTT in HCC patients. Please see result section page 11 for more details too.

Related figure 4 legend: (Q) High BMI1 and CTSB expression strongly correlated with poor overall survival in HCC patients as compared to patients with low BMI1 and CTSB expression (n= 194 HCC patients, our cohort).

Related supplement figure 4 legend: (J) Kaplan-Meier survival study of HCC patients with either high or low CTSB expression together with or without BDTT (n=194 HCC patients, our cohort).

- Fig. 5: Ctrl and scramble columns should display SD bar.

We thank for the reviewer's comment. In figure 5, we show the results after normalized to the control/scramble cells. Therefore, the ctrl and scramble columns do not have error bars after the normalization. It is done as described before in the study of Barczak et al., Nature Comms 2023 and Rho et al., Cell metabolism 2023.

Minor comments:

- In line 135 authors describe findings on TIC-related gene signature however, Fig. 1G and H refer to protein expression detected by IHC and only Fig. S1C refers to gene signature. This should be amended on the text.

We are sorry for the mistake. We have now amended all these terms throughout the text, figure and figure legend. Please see the revised figures and figure legends and supplementary figure legends.

- Fig 1D. legend is incorrect according to the text.

We are sorry for the mistake. We have now amended it in the figure legend. Please see the figure legend page 36.

- Fig 4B: extracellular region is not displayed in the figure in red as a cellular process related to CtsB in the figure. BP, CC, MF meaning of should be stated in figure legend.

We apologize for the mistake. We have now made these changes in the figure 4B and its legend as well as the main text. Please see result section page 11 and figure 4B legend page 39.

- Fig 6G: It is not clear whether the ELISA was performed in media or cell lysate.

We have now made it clear in our result and figure legend sections. The ELISA was performed in media. Please see the figure 6G legend page 41.

- Fig 6G and I: X axis are not scaled properly and looked distorted.

We apologize for the mistake. We have now adjusted this problem. Please see new figure 6G and I.

- Manuscript should be revised for some typos such as “corelated” in line 350.

We apologize for the typos. We have now made the corrections throughout the text.

Reviewer #4 - HCC thrombosis (Remarks to the Author):

In this manuscript, Xu et al explore how bile duct tumor thrombus (BDTT) develops in HCC. Using HCC patient samples, they find that the incidence of bile duct tumor thrombus is higher in cancers with greater BMI1 expression. Overexpression of BMI1 in different cell lines induced proliferation, migration, and tumor formation in subcutaneous injections. Injection of BMI1 overexpressing cells into Glisson's capsule lead to tumor formation and the presence of jaundice with BDTT at a rate of 40%. Using proteomics, the authors found that the protein level of CTSB was increased with BMI1 overexpression in vitro. Using CTSB targeted siRNAs or small molecule inhibitors, the authors could reduce in vitro invasion and prevent the development of jaundice and BDTT. Furthermore, informatic and ChIP identified miR-218-1-3p as repressed by BMI1 and this miR appeared to bind to the 3' UTR of CTSB. The authors found that injection of miR-218-1-3p into their BDTT model could reduce the BDTT. While the formation of BDTT is somewhat low in their model; which could reduce its generalized use, the mechanism of BMI1/miR-218-1-3p/CTSB/ they describe is interesting and relevant to those who study HCC. I do have a few comments that I believe would help strengthen the manuscript and improve its clarity.

We appreciate the reviewer's keen interest in our recently uncovered BDTT molecular mechanism. It is worth noting that, until now, there has been a notable absence of a suitable BDTT animal model within the realm of HCC research. To address this gap, we have successfully developed the inaugural HCC animal model boasting a considerable incidence rate of BDTT. This model has equipped us with a valuable tool to delve into the underlying mechanisms and potential therapeutic targets pertinent to advanced HCC cases featuring BDTT. Your recognition of our efforts is genuinely encouraging, and we remain open to further discussions and insights.

1. There are multiple grammatical errors and portions of writing that are of poor English. These should be corrected.

We apologize for the grammatical errors and poor English writing in some sections. Our manuscript has been now proof-read by a naïve English speaker.

2. It was not clear to me how it was determined what constituted a BMI1High and BMI1Low tumor sample. In Figure 1A, the low sample seems to have no expression by IHC. Was this distinction performed based on IHC or RNA level? What cutoff was used?

We would like to thank for the reviewer's comment. For your interest, we have now replaced the representative IHC picture of BMI1 staining in HCC patient with low BMI1 expression in figure 1A with a better representative IHC image (please see new figure 1A). We also make it more clear about our selection process in the method section (please see page 21 and 26). In this study, the median BMI1/CSTB mRNA expression level/staining intensity was used as a cutoff value for stratifying patients into high or low BMI1/CSTB expression group. Please see method section page 21 and 26.

3. Bile duct tumor thrombus is a relatively rare complication of HCC; yet, in the database from the authors there appears to be a high proportion of these patients (43 out of 194, Fig 1D). Is there a reason for this? It would be helpful to have the clinical characteristics (labs, demographics, etc) of these patients divided by the presence and absence of BDTT and explained how they were determined to have BDTT. Additionally, what is the total size of this cohort as in some survival

curve figures there are less patients and some there are more patients. Is there a reason that some patients are being excluded?

We would like to thank for the reviewer's comment. We have provided the clinical characteristics of our patients in supplementary table 1. We have also provided more explanation about the collection of our patient cohort in the method section (please refer to page 21). The research collected in this study involves HCC patients who underwent surgical resection at Sun Yat-sen Memorial Hospital of Sun Yat-sen University between 2012 and 2018. During this period, a total of 1791 cases of HCC patients underwent liver cancer resection surgery. Among them, there were 53 cases of patients with combined bile duct cancer thrombi, and the incidence rate of bile duct cancer thrombi was 2.96%, consistent with relevant literature reports (Qiao et al., BMC Gastroenterol. 2016; 16: 11; Shiomi M et al., Surgery. 2001;129(6):692–8). The inclusion criteria for this study were: 1. Pathological diagnosis of liver cancer; 2. R0 resection; 3. Complete clinical data and follow-up information; 4. Specimens available. Exclusion criteria included: 1. Preoperative radiotherapy or chemotherapy; 2. Concurrent with other tumors. Among them, the diagnostic criteria for HCC patients with combined bile duct cancer thrombi were: observation of tumor foci within the bile duct lumen under the microscope, with pathological diagnosis of HCC combined with bile duct cancer thrombi. Based on these selection criteria, 151 cases of HCC patients without bile duct cancer thrombi and 43 cases of HCC patients with combined bile duct cancer thrombi were included in this study. The study was conducted continuously with these 194 cases of HCC patients, without any deletions. Please see method section page 21 for more details.

4. Figure 4Q: Do patients have tumors that are BMI1high/CTSBlow and BMI1low/CTSBhigh and what happens to them?

In accordance with the valuable suggestion from reviewer 3, we initially made the decision to exclude the clinical data of patients with BMI1high/CTSBlow and BMI1low/CTSBhigh from Figure 4Q. This precautionary step aimed to prevent any potential confusion that could arise from the analysis. However, in light of your feedback, we have subsequently reintegrated the data pertaining to patients with BMI1high/CTSBlow and BMI1low/CTSBhigh into Figure 4Q. Our meticulous data analysis has yielded a notable finding: HCC patients presenting with high BMI1 and low CTSB expression do not exhibit a significant difference in overall survival in comparison to patients demonstrating low BMI1 and high CTSB expression. This observation diverges from our previously established results, where HCC patients with high BMI1 and CTSB expression displayed poorer overall survival than those with low BMI1 and CTSB expression. Please refer to the revised Figure 4Q and figure below alongside its corresponding legend in the manuscript as well as result section page 12.

Related figure 4 legend: (Q) High BMI1 and CTSB expression strongly correlated with poor overall survival in HCC patients as compared to patients with low BMI1 and CTSB expression (n= 194 HCC patients, our cohort).

5. It would be informative if the authors could label their WBBMI1 cells with a fluorescent reporter to see if the breaks in the bile duct wall are due to tumor invasion vs immune cells as their H and E stains seem to indicate a marked inflammatory presence. Alternatively, if the authors have a specific antibody for their cells, immunostaining could be used.

We extend our appreciation for the valuable input provided by the reviewer. Acting upon the recommendation, we have conducted additional co-immunostaining experiments, which clearly demonstrate that BMI1-overexpressing cells invade the CK7 (a bile duct epithelial cell marker)-positive bile ducts, leading to the formation of BDTT (as shown in the new Supplementary Figure 3A). We have also executed the orthotopic injection of GFP-expressing WB^{ctrl} or WB^{BMI1} cells into the livers of nude mice. Subsequently, the tumors obtained from WB^{BMI1} implanted mice were meticulously fixed and subjected to co-immunostaining using GFP and CK7 antibodies. Our latest findings unequivocally validate that the GFP-overexpressing WB^{BMI1} cells invaded from the primary tumor site into the bile ducts, thereby instigating the formation of tumor thrombi (refer to supplementary figure 3B). For a visual representation of these conclusive results, we invite you to explore the newly incorporated supplementary figure 3A and B, and figure below as well as result section page 9-10 and discussion section page 18.

Related supplementary 3 figure legend: (A) Representative immunofluorescent images of BMI1 and CK7 stained tumor sections derived from WB^{ctrl} or WB^{BMI1} implanted mice are given. (B) Provided are representative immunofluorescent images capturing the GFP overexpressing BMI1 cells and CK7 antibody staining within tumor sections extracted from nude mice implanted with GFP labelled WB^{ctrl} or WB^{BMI1} cells.

Reviewers' Comments:

Reviewer #1:

Remarks to the Author:

The authors have convincingly addressed the concerns raised by the reviewer.

Reviewer #2:

Remarks to the Author:

Appreciate the extensive revision. I am happy with the revised version and have no further questions.

Reviewer #3:

Remarks to the Author:

The authors have addressed all my questions and comments. I have no further comments.

Reviewer #4:

Remarks to the Author:

The authors have satisfactorily addressed my concerns, no further comments.

REVIEWERS' COMMENTS

Reviewer #1 (Remarks to the Author):

The authors have convincingly addressed the concerns raised by the reviewer.

Comment: We appreciate the reviewer's suggestions, which have helped us improve our manuscript. We are pleased to know that he/she believe we have convincingly addressed the concerns he/she raised.

Reviewer #2 (Remarks to the Author):

Appreciate the extensive revision. I am happy with the revised version and have no further questions.

Comment: We're glad to hear that you are satisfied with the revised version of the manuscript.

Reviewer #3 (Remarks to the Author):

The authors have addressed all my questions and comments. I have no further comments.

Comment: We are pleased to hear that all your questions and comments have been adequately addressed.

Reviewer #4 (Remarks to the Author):

The authors have satisfactorily addressed my concerns, no further comments.

Comment: We appreciate the reviewer's feedback, and we're glad to hear that your concerns have been satisfactorily addressed.